# Cassiosomes are stinging-cell structures in the mucus of the upside-down jellyfish *Cassiopea xamachana*

Cheryl L. Ames[1,2,3,12]*, Anna M.L. Klompen [3,4,12], Krishna Badhiwala[5], Kade Muffett[3,6], Abigail J. Reft[3,7], Mehr Kumar[3,8], Jennie D. Janssen [3,9], Janna N. Schultzhaus[1], Lauren D. Field[1], Megan E. Muroski[1], Nick Bezio[3,10], Jacob T. Robinson[5], Dagmar H. Leary[11], Paulyn Cartwright[4], Allen G. Collins [3,7] & Gary J. Vora[11]*

Snorkelers in mangrove forest waters inhabited by the upside-down jellyfish *Cassiopea xamachana* report discomfort due to a sensation known as stinging water, the cause of which is unknown. Using a combination of histology, microscopy, microfluidics, videography, molecular biology, and mass spectrometry-based proteomics, we describe *C. xamachana* stinging-cell structures that we term cassiosomes. These structures are released within *C. xamachana* mucus and are capable of killing prey. Cassiosomes consist of an outer epithelial layer mainly composed of nematocytes surrounding a core filled by endosymbiotic dinoflagellates hosted within amoebocytes and presumptive mesoglea. Furthermore, we report cassiosome structures in four additional jellyfish species in the same taxonomic group as *C. xamachana* (Class Scyphozoa; Order Rhizostomeae), categorized as either motile (ciliated) or nonmotile types. This inaugural study provides a qualitative assessment of the stinging contents of *C. xamachana* mucus and implicates mucus containing cassiosomes and free intact nematocytes as the cause of stinging water.

[1] National Academy of Sciences, National Research Council, Postdoctoral Research Associate, US Naval Research Laboratory, Washington, DC 20375, USA. [2] Graduate School of Agricultural Science, Tohoku University, Sendai 980-8572, Japan. [3] Department of Invertebrate Zoology, National Museum of Natural History, Smithsonian Institution, Washington, DC 20560, USA. [4] Department of Ecology and Evolutionary Biology, University of Kansas, Lawrence, KS 66049, USA. [5] Department of Bioengineering, Rice University, Houston, TX 77005, USA. [6] Texas A&M University at Galveston, Galveston, TX 77553, USA. [7] National Systematics Laboratory of the National Oceanic Atmospheric Administration Fisheries Service, National Museum of Natural History, Smithsonian Institution, Washington, DC 20560, USA. [8] Stanford University, Stanford, CA 94305-2004, USA. [9] National Aquarium, Baltimore, MD 21202, USA. [10] California State University, Monterey Bay, CA 93955, USA. [11] Center for Bio/Molecular Science and Engineering, US Naval Research Laboratory, Washington, DC 20375, USA. [12] These authors contributed equally: Cheryl L. Ames, Anna M. L. Klompen. *email: ames.cheryl.lynn.a1@tohoku.ac.jp; gary.vora@nrl.navy.mil

Jellyfish, along with corals, anemones, hydroids, and myxozoans, belong to the phylum Cnidaria, the earliest diverging venomous animal lineage[1,2]. These diploblastic animals have two so-called epithelial layers, outer ectoderm and inner endoderm, separated by a gelatinous extracellular matrix called mesoglea[3,4]. Despite their seemingly simple morphology, cnidarians have adapted globally to most saltwater habitats and some freshwater environments[1,5]. As such, cnidarians have evolved a remarkable envenomation mechanism that involves the deployment of subcellular stinging capsules called nematocysts from cnidarian-specific cells called nematocytes, which vary in size, morphology, and bioactive contents[6–8]. Sea anemones possess unique nematocyte-rich structures (e.g., acrorhagi, acontia)[9,10] and employ strategies such as tentacle and column contraction and expansion to enhance nematocyst deployment for prey capture and protection, while in medusae (i.e., jellyfish) the first line of defense is their extendable nematocyte-laden tentacles that envenomate prey and predators they encounter in the water column[1], as well as humans participating in marine recreation. In addition to direct stings caused by jellyfish, indirect stings have also been reported. Some possible explanations for indirect jellyfish stings are contact with tentacle fragments in the water (e.g., jellyfish stings in offshore fishers[11]), envenomation by juvenile venomous jellyfish (e.g., Irukandji-like syndrome in United States Military combat divers[12]) or Sea Bathers Eruption caused by microscopic jellyfish life forms (e.g., *Linuche unguiculata*[13]).

Another indirect stinging mechanism is through mucus, such as in medusae of the upside-down mangrove jellyfish *Cassiopea xamachana* Bigelow 1892 (Class Scyphozoa; Order Rhizostomeae), an emerging cnidarian model for its relevance to the study of coevolution as well as symbiosis-driven development (reviewed in ref. [14]). Additionally, the ubiquity of *Cassiopea* medusae in healthy mangroves has earned the upside-down jellyfish status as a potential bioindicator species for coastal management and conservation efforts[15,16]. *Cassiopea* is known to release large amounts of mucus into the water column[17–24], which has been referred to as toxic mucus due to reports of nematocysts found freely suspended in the vicious substance[17,20,25]. For instance, *Cassiopea* mucus is known to kill certain species of fish on contact[24]. *Cassiopea* is an exception to the iconic image of a jellyfish in that it lacks marginal tentacles and, instead of swimming in the water column, lies apex-down on the substrate in mangrove forests, seagrass beds or other coastal waters with its relatively short oral arms facing upward (reviewed in ref. [26]). Despite this benthic lifestyle, warnings have been published alerting sea bathers of the stinging water or toxic mucus phenomena blamed on unidentified potent little grenades in the water column surrounding *Cassiopea* medusae[20,21]. In general, *Cassiopea* stings are categorized as mild to moderate in humans, but crude venom extracted from the nematocysts displays hemolytic, cardiotoxic and dermonecrotic properties[27–30], suggesting that excessive exposure may be detrimental for humans.

During the course of this study, a review of the old literature on *Cassiopea* revealed a probable explanation for the grenades reported in stinging water. First, Perkins[17] discovered in the mucus of *Cassiopea* undeployed nematocysts and ciliated innumerable minute spherical bodies, the latter of which were dismissed as non-coelenterate (i.e., non-cnidarian) in nature. Next, a brief description was published by Smith[40] of peculiar structures found in the oral vesicles (i.e., vesicular appendages) of the oral arms of *C. xamachana* and conspecific *C. frondosa* medusae that were 'shot' at prey, which he dubbed small bags of mesoglea and nematocysts and suggested might play a role in predation. Finally, Larson[43] reported polygonal-shaped bodies on the flatted sides of the oral vesicles (i.e., vesicular appendages) of the oral arms of *C. xamachana* and *C. frondosa* corresponding to nematocyst clusters

that released upon contact, to which he attributed a role in defense. The sum of these reports suggests that an investigation of the contents of *Cassiopea* mucus is needed to test the hypothesis that undeployed nematocysts and/or another nematocyst-bearing structure(s) present within the mucus of the upside-down jellyfish together are responsible for the phenomenon of stinging water experienced by humans in the vicinity of *Cassiopea* medusae.

In this study, we used a combination of microscopy, microfluidic devices, molecular biology techniques, mass spectrometry-based proteomics, and other experimental assays to provide the first detailed description, to our knowledge, of the contents of the mucus liberated from lab-reared *Cassiopea xamachana* medusae. Released within the mucus, we discovered three types of undeployed nematocysts, as well as microscopic, motile, cellular masses composed of nematocytes that we formally call cassiosomes. While cassiosomes bear some resemblance to another cnidarian structure originating in mesenteries of the starlet sea anemone *Nematostella* called nematosomes[31], the unique traits of cassiosomes in *C. xamachana* include their release into the water column within mucus, the ability to trap and kill prey as mobile grenades outside of the medusa, their organization as an outer epithelial layer surrounding a mostly empty core (rather than a solid ball of cells), and the presence of centrally-located endosymbiotic *Symbiodinium* dinoflagellates. We document the presence of cassiosomes in five species spanning four families of the order Rhizostomeae, while also confirming their absence in the moon jellyfish *Aurelia* (Semaeostomeae), a representative of the sister lineage, and discuss the possibility of a single evolutionary event behind this envenomation strategy which, to our knowledge, is unique. Despite the growing body of work on *C. xamachana* from an organismal biology perspective[14,26], this study is the first to directly investigate stinging properties of the mucus of this jellyfish and the potential ecological and evolutionary relevance.

## Results

**Study overview.** Observations were made on lab-reared *Cassiopea xamachana* (see the "Methods" section) and on medusae in their natural habitat in waters of Florida Keys mangrove forests (Fig. 1a, b). In both cases, medusae were observed releasing copious amounts of mucus into the water when surrounding water was disturbed (by jellyfish aquarists and/or snorkelers), or when prey items were provided (e.g., *Artemia* nauplii in aquarium-reared medusae). Stinging water phenomenon was experienced by the authors while handling lab-reared and/or wild *C. xamachana* and other rhizostome jellyfish examined in this study (species list provided below).

***Cassiopea xamachana* overview. Life cycle and endosymbiosis** *C. xamachana* medusae start out like most scyphozoan jellyfish, as an asexual microscopic polyp that metamorphosizes into a sexually reproducing medusa via a process known as strobilation[19]. However, they differ from most jellyfish in that they host endosymbiotic dinoflagellates (also called zooxanthellae)[26,32]. Colonization of polyps by algal endosymbionts is the most common type of intracellular mutualism among cnidarians of the class Anthozoa (e.g., corals and anemones), and although it is less common in jellyfish species[33], endosymbiosis triggers the start of *C. xamachana* polyp strobilation[18,19]. During the sessile life stage, these polyps engulf dinoflagellates (unicellular algae called *Symbiodinium*) via the manubrium (feeding tube), which are then phagocytosed by endodermal cells[18,33]. Bound by a membrane complex that combines host and infecting cell membranes (called a symbiosome[33]), *Symbiodinium* spp. migrate to the polyp mesoglea and remain there housed in endodermal cells,

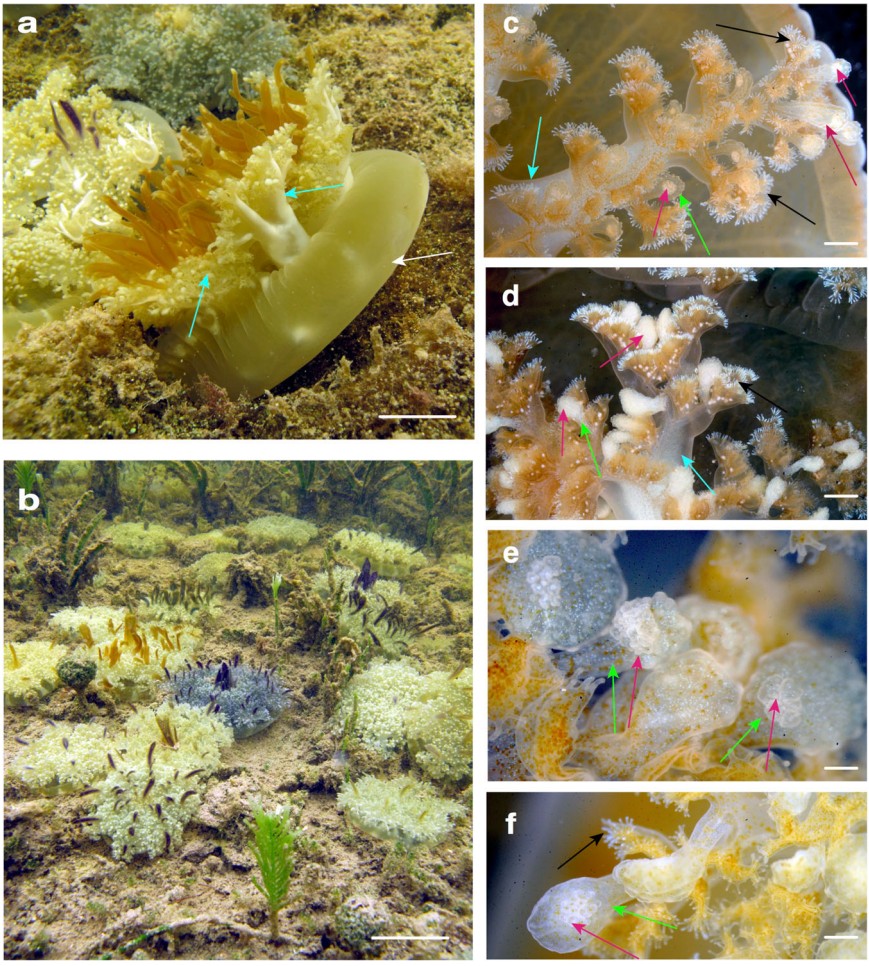

**Fig. 1 Medusae of the upside-down mangrove jellyfish *Cassiopea xamachana*. a, b** *C. xamachana* medusae (5–12 cm diameter) resting on umbrella apex (white arrow) with oral arms (cyan arrows) facing up, observed by authors in the natural mangrove habitat in Key Largo, Florida (USA). Images courtesy of A. Morandini. **c–f** Cassiosome nests (pink arrows) observed as white bulging spots at the termini of vesicular appendages (green arrows) off-branching from areas of frilly digitate cirri (black arrows) on medusa oral arms (cyan arrows). Some cassiosome nests appear less full than others. Scale bars: **a** = 2 cm; **b** = 5 cm; **c, d** = 1 mm; **e, f** = 0.5 mm.

transformed into amoebocytes[34]. Shortly after infection with endosymbionts, *C. xamachana* polyps undergo strobilation, and the apical portion metamorphosis into an ephyra (juvenile medusa) which then develops into a sexually mature male or female medusa, with multiple color variants based on endosymbionts[19,26] (Figs. 1a, b, 2a). *Symbiodinium*-generated photosynthates support the jellyfish host metabolism, growth, reproduction and survival. This promotes conservation and recycling of essential nutrients, given their strategic presence amidst downwelling light, which is of unrivaled ecological importance for coral reefs and *Cassiopea* populations alike[26,32,33].

**Medusa feeding**. Feeding studies on *Cassiopea* medusae show that prey capture occurs as a result of perpetual medusa pulsation that carries the prey into the subumbrellar space and then onto the oral arms where they are held by nematocyst-rich digitate fringed lips and vesicular appendages (i.e., small oral vesicles) (Fig. 1a–f), eventually being reduced to fragments. Finally, food particles are then forced into the oral ostia of secondary mouths, and ingested via ciliary action. *Cassiopea* are opportunistic predators, feeding on a broad range of prey items (e.g., crustaceans, nematodes, eggs) in the field (Larson[43]), while in the lab polyps and medusae are fed *Artemia salina* (1–3 days old, lab reared).

**Cassiosomes overview**. *Cassiosomes morphology*: Numerous, motile cellular structures, which we call cassiosomes, were observed suspended within mucus released by *C. xamachana* medusae (3.0–8.8 cm umbrella diameter) (Fig. 2a–f) in response to feeding or mild disruption with short bursts of seawater from a pipette. Herein, we describe cassiosomes in *C. xamachana* as microscopic (100–550 μm in diameter), irregularly-shaped cellular masses whose peripheral cell layer is primarily composed of nematocytes and other irregular ectodermal cells that surrounds a space containing amoebocytes —some hosting *Symbiodinium* and others lacking them—among presumptive mesoglea (Fig. 2f).

*Cassiosome motility*: When multiple *C. xamachana* medusae were placed together and agitated by directing water at their oral arms using a glass pipette, they consistently released cassiosome-laden mucus within 5–10 min (Fig. 2a–c) for periods lasting several hours. When collected mucus was transferred to a small glass dish, cassiosomes moved around within the mucus for about 15 min and then descended to the bottom of the dish, leaving the neutrally buoyant mucus. This permitted efficient isolation of numerous cassiosomes (Fig. 2b–f), which remained in constant motion by rotating and displacing in various directions along the bottom, but never elevating from the bottom of the dish (Supplementary Movie 1). Isolated cassiosomes remained motile for up to 10 days, gradually losing their corrugated appearance

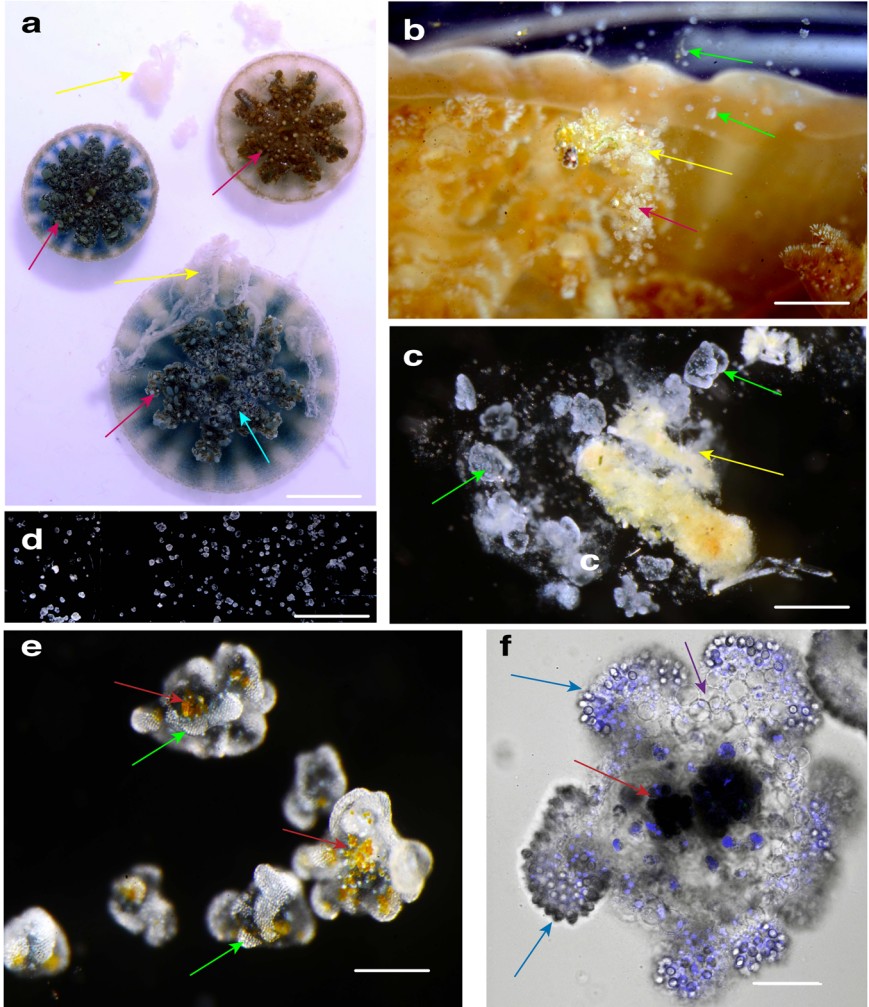

**Fig. 2 Observations of mucus and cassiosome release in _Cassiopea xamachana_ medusae. a** _C. xamachana_ releasing mucus (yellow arrows) following collection in the field by authors (Bonaire, The Netherlands). Cassiosome nests (pink arrows) appear as light bulging spots at the termini of vesicular appendages (cyan arrow). **b** Mucus (yellow arrow) released into the water by _C. xamachana_ in the lab—small white flecks correspond to live cassiosomes (green arrows). **c** Live cassiosomes (green arrows) suspended in mucus (yellow arrow) harvested after release from _C. xamachana_. **d** Multiple motile cassiosomes isolated from _C. xamachana_ mucus. **e** Live cassiosome close-up (green arrows), showing irregular shape and centralized _Symbiodinium_ dinoflagellates as amber spheres (red arrows). **f** Confocal image of highly motile cassiosome immobilized on MatTek glass bottom dishes coated with Cell Tak adhesive (Corning); image collected with ×60 objective (oil) reveals organization of the peripheral cell layer: NucBlue-Hoescht 33342 (1,100) (ThermoFisher) stains nuclei (blue) of nematocytes with peripheral nematocytes bearing O-isorhiza nematocysts (blue arrows, DIC) and non-nematocyte ectodermal cells (purple). DIC shows centrally _Symbiodinium_ (dark spheres, red arrows) occupy presumptive _C. xamachana_ amoebocytes in an otherwise acellular core. Scale bars: **a** = 3 cm; **b** = 3 mm, **c** = 1 mm;, **d** = 5 mm; **e** = 300 μm, **f** = 50 μm.

after 5 or 6 days and shrinking in size to a smooth spherical shape until movement ceased and cassiosomes disintegrated. Additionally, custom-designed microfluidic devices with channels equal to or slightly bigger than the cassiosomes[35] (Supplementary Fig. 1d) were used to observe cassiosomes of _C. xamachana_ to gain a better understanding of their motility, and three-dimensional irregular, popcorn-shaped structure. These microscopic observations revealed motile cilia extending from the periphery that propel cassiosomes.

_Cassiosomes kill prey_: We conducted assays to determine if cassiosomes were capable of trapping _Artemia_ nauplii provided as food in aquarium-reared medusae since so-called non-penetrant O-isorhiza nematocysts are the only type found in cassiosomes of _C. xamachana_. For this purpose, microfluidic chambers[35] provided an arena in which to document immobilization and rapid trapping of _Artemia_, which were killed almost immediately upon substantial contact with cassiosomes (Fig. 3a–c) (Supplementary Movie 5).

When isolated cassiosomes were added to dishes (150 ml) containing abundant _Artemia_ nauplii (Supplementary Movie 2), cassiosomes that encountered the underside of the nauplii carapace immediately immobilized and killed the prey items (Fig. 3a–c). In cases where _Artemia_ nauplii came into contact only briefly with cassiosomes, prey were able to escape rapid immobilization and death (Supplementary Movie 5). Furthermore, during discharge assays when either FASW (Filtered Artificial Seawater) or mucus containing no cassiosomes, following manual removal, was added (see the "Methods" section), _Artemia_ were not affected and continued to swim around in the dish (Supplementary Movies 4 and 3).

_Cassiosome ultrastructure_: In order to better understand their organization, live cassiosomes were isolated from _C. xamachana_ mucus. After fixing, dehydrated specimens were examined using scanning electron microscopy (SEM) (Fig. 4a–f). The cassiosome perimeter was found to be lined with nematocyst capsules and

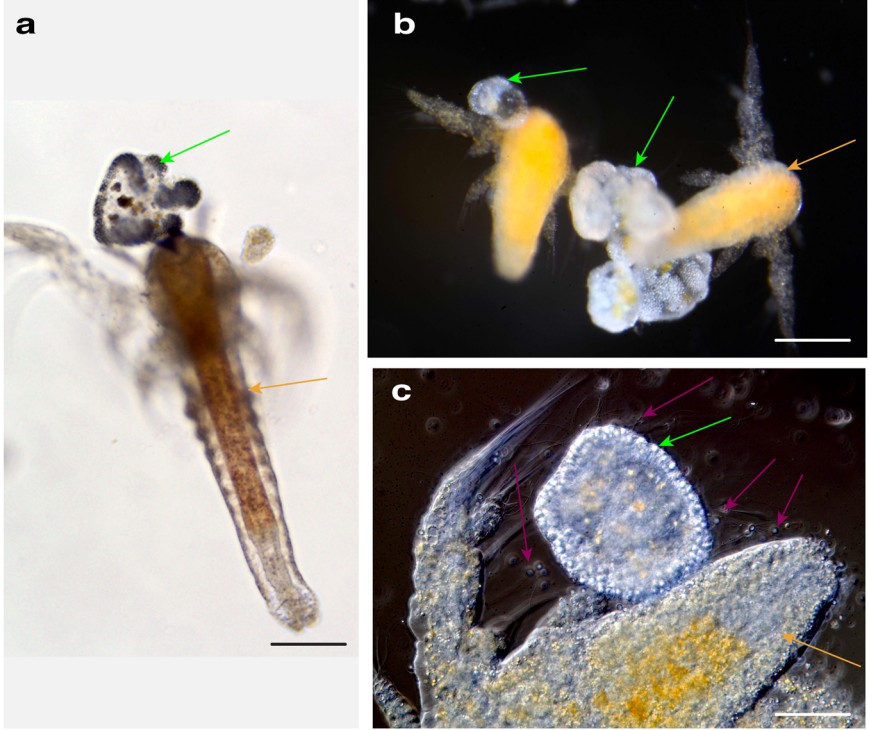

**Fig. 3 Cassiosomes are capable of killing brine shrimp. a** Dead 2-day old *Artemia* nauplii (orange arrow) with cassiosome (green arrow) attached to carapace, imaged within a microfluidic chamber. **b** Cassiosomes (green arrow) lodged into two different 1-day old *Artemia* nauplii (orange arrow). **c** 1-day old *Artemia* nauplii (orange arrow), immobilized following cassiosome (green arrow) attachment to the carapace with visibly discharged nematocysts (fuchsia arrows). Scale bars = 200 μm.

numerous long, spiny tubules (Fig. 4a–f) extruded from abundant O-isorhiza nematocysts in the periphery following spontaneous deployment (likely during the dehydration process (see the "Methods" section)). More abundant on the surface, however, were much thinner filaments corresponding to abundant cilia connected to ectodermal cells (Fig. 4a–c). Close observation of several cassiosomes via SEM revealed along the outer layer emptied out regions appearing as collapsed cell membrane remnants of deployed nematocysts (Fig. 4a–f), underneath which could be seen an amorphous thick central extracellular matrix-like substance (Fig. 4e, f). Confocal microscopy on fixed cassiosomes with labeled nuclei, nematocysts and cilia (see the "Methods" section) (Fig. 5a–d) corroborated these findings of an organized cell mass. Cassiosomes are composed of a peripheral layer of nematocytes bearing O-isorhiza nematocysts (Fig. 5a) patterned with presumptive ectoderm cells that lack nematocysts, from which numerous cilia protrude (Fig. 5a–d). This outer layer surrounds centralized clusters of *Symbiodinium* endosymbionts (i.e., hosted by amoebocytes) within an otherwise apparently acellular region (Fig. 4e, f).

**Cnidome of *C. xamachana*.** The term cnidome refers to the dynamic repertoire of nematocyst types in a cnidarian species[7,8]. The cnidome, a species-specific trait, often changes throughout the life cycle of the jellyfish as it undergoes metamorphosis from a sessile polyp, to strobila, and then to juvenile and sexually mature medusa. Given reports implicating nematocysts, or tiny little grenades, within *Cassiopea* mucus as the cause of stinging water, we sought to characterize the cnidome of this species at several life stages, and within the cassiosomes and contents of the mucus.

Nematocyst measurements were plotted for the following life stages of *C. xamachana*: polyps (*n* = 3); strobilating/released ephyrae (*n* = 3) and medusae (2.4–8.8 cm in diameter (*n* = 3));

mucus; and cassiosomes (isolated from mucus) (Fig. 6a–d). Measurements of undischarged nematocysts of each type in the corresponding subsample revealed that O-isorhizas nematocysts are absent in polyps, but appear in medusae from the onset of ephyra development during strobilation (Fig. 6; Supplementary Table 1). We also observed penetrant nematocytes, birhopaloids and euryteles, which cannot be distinguished in the undeployed state (intact) within *C. xamachana* tissue using light microscopy. Therefore in this study, these two nematocyst types were analyzed together as rhopaloids (Fig. 6a–d) (as per ref. [36] in *C. andromeda*); hence, rhopaloids account for a larger proportion of the cnidome in the medusa and mucus than distinct isorhiza types. An assessment of the inventory of nematocysts freely suspended within the mucus yielded a similar nematocyst profile to that of the medusa (Fig. 6a), albeit with a proportionately higher number of rhopaloids, which are implicated in envenomation[28,29]. Conversely, isolated cassiosomes of *C. xamachana* contain exclusively O-isorhiza nematocysts which are a ubiquitous type in jellyfish tentacles, functioning in prey capture and predation.

**C. xamachana toxin proteins in cassiosomes.** Over a century ago, Perkins[17] documented that disturbed *C. xamachana* medusae produced mucus containing ciliated structures as innumerable minute spherical bodies containing unicellular zooxanthellae within the interior, which he considered to be parasitic larvae, and claimed it was "impossible to regard [these structures] as of coelenterate [Cnidaria] affinities". These details suggest that Perkins observed what we have herein identified as cassiosomes, but mistook them for entirely unique, non-cnidarian organisms. In order to test this theory, and properly classify cassiosomes as belonging to *C. xamachana*, rather than being unknown organism, we used real-time PCR (qPCR) assays to target species-specific cnidarian toxins, employing three custom-designed

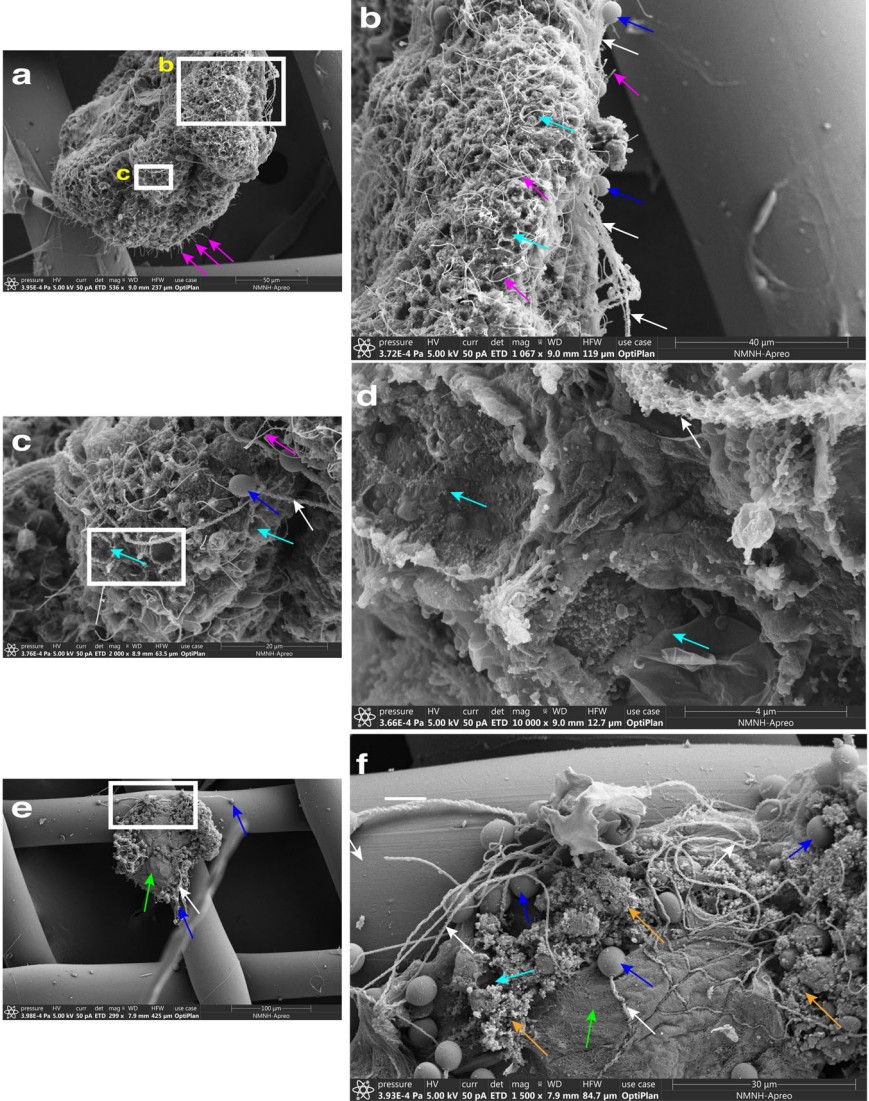

**Fig. 4 Cassiosome organization revealed via SEM. a** An individual cassiosome poised in a 100-µm mesh opening, revealing the irregular 'popcorn' shape of the cassiosome bearing numerous cilia (pink arrows) protruding from the peripheral layer. White rectangles correspond to magnified region shown in (**b**) and (**c**). **b, c** Close up of the cassiosome reveals cilia (pink arrows), and thicker tubules (white arrows) of discharged O-isorhiza nematocyst capsules (blue arrows) in the periphery which is lined with collapsed nematocytes (cyan arrows). Spiny nematocyst tubules (white arrows) are outnumbered by the abundant cilia (thinner filaments) (pink arrows). **d** Depressions fringed by deflated cell membranes outlining nematocytes (cyan arrows in **c** and **d**) following deployment of O-isorhiza nematocysts (blue arrows in **c**) along the cassiosome surface. **e** A different cassiosome from that in (**a–d**). All cellular components were lost (possibly in dehydration stage of methods) but for discharged O-isorhiza nematocyst capsules (blue arrows) and corresponding spiny tubules (white arrows) in the cassiosome periphery lined with collapsed nematocytes (cyan arrows). White rectangle corresponds to magnified region shown in (**f**) revealing the dense central, fibrous extra-cellular matrix (green arrow). Unidentified microscopic particles also present (orange arrows). Scale bars as indicated in SEM images.

primer pairs (Supplementary Fig. 1e–g) that we designed from the publicly available *C. xamachana* genome (see Data Availability section).

We targeted a cnidarian-restricted CrTX/CaTX family toxin[37,38] gene in DNA extracted separately from *C. xamachana* medusa tissue and isolated cassiosomes, and also from tissue of the jellyfish *Aurelia sp.* (Class Scyphozoa) and a more divergent jellyfish species *Alatina alata* (Class Cubozoa) for comparison. Amplification of the qPCR gene target was observed for both *C. xamachana* tissue and cassiosome samples using primers for the CrTX/CaTX gene, herein named CassTX-C (see Data Availability section below). Conversely, failure to amplify the target in non-*Cassiopea* medusozoans used in this study (despite reports of CrTX/CaTX genes documented in both *Aurelia* sp

and *A. alata*[14,39]) validates the specificity of our primers to a *C. xamachana*-derived gene target, indicating that cassiosomes originate in *Cassiopea* medusae.

To validate the potential for envenomation by cassiosomes, rather than solely by suspended intact nematocysts in the mucus released by medusae, we used LC-MS/MS analyses to confirm the presence of the same CrTX/CaTX toxin proteins in two *C. xamachana* sample types: cassiosomes isolated from mucus released from ~20 medusae over a 7-h period, and several vesicular appendages containing cassiosome nests, dissected from multiple medusae (e.g., Fig. 1c–f). A shotgun proteomic analysis identified three isoforms of the target toxin family encoded in the *C. xamachana* genome[14], which we call CassTX-A, CassTX-B and CassTX-C (Fig. 7). Each toxin protein was identified with

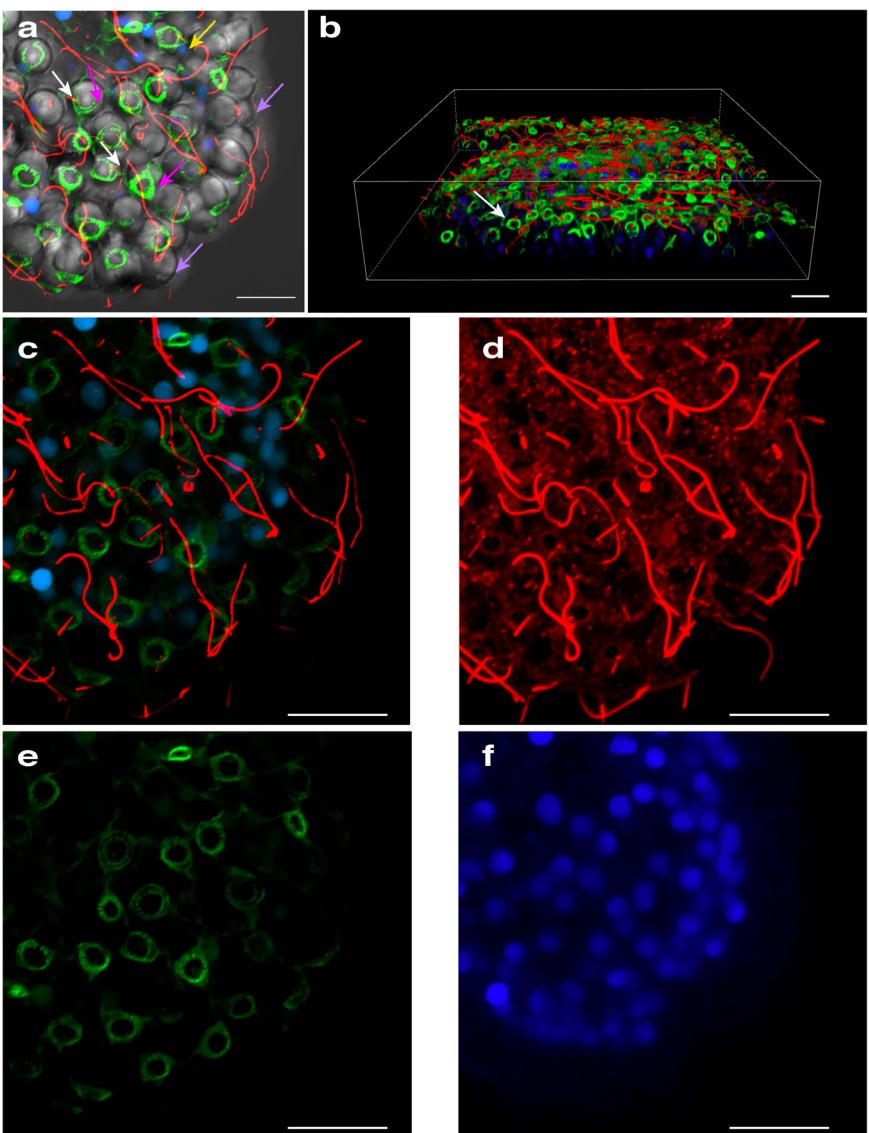

**Fig. 5 Characterization of cassiosome ultrastructure. a–f** Individual cassiosome fixed and labeled with Tubulin Antibody, ActinGreenTM and NucBlueTM, and mounted in 80% glycerol in PBS on glass slides for imaging. Imaging was performed with both DIC and confocal laser scanning with lines at 405, 488, 561, and 640 nm, and collected with a Plan Apo ×100 objective. **a** DIC reveals peripheral layer of nematocytes bearing spherical O-isorhiza nematocysts (lavender arrows). Tubulin (red) reveals cnidocils (short filaments marked by white arrows) extending from apex of nematocytes, and motile cilia (long filaments) originating from non-nematocytes ectoderm cells organized in patches along the peripheral layer among nematocyte-rich areas. NucBlue (blue) reveals nuclei (yellow arrows) of peripheral epithelial layer – nematocytes and other ciliated ectoderm cells. Actin (green) reveals actin basket (pink arrows) formed around the apex of nematocysts. **b** 3-D construction of Z-stack magnified confocal images corresponding to (**a**) and (**c–f**) shows tubulin (red) of cnidocils (short filaments marked by white arrows) extending from around apex of nematocytes and motile cilia (long filaments) originating from non-nematocytes putative ectoderm cells organized in patches along the peripheral layer among nematocyte-rich areas. NucBlue (blue) reveals nuclei of peripheral epithelial layer—nematocytes and other ciliated ectoderm cells. Actin (green) reveals actin basket forming around the apex of nematocysts. Scale bars = 10 μm.

multiple unique peptides and ≥17.0% protein coverage (Fig. 7; Supplementary Fig. 3; Supplementary Table 2), with the exception of CassTX-C, which in the cassiosomes was not assigned with sufficient confidence (based on Mascot Score). Peptides identified and aligned to the three *Cassiopea* toxin homologs are shown in Supplementary Fig. 3, along with the protein gel (Supplementary Fig. 4).

**Cassiosome provenance and development**. A visual inspection of *C. xamachana* oral arms during mucus release revealed that cassiosomes occur as warty clusters within a shallow pocket on

the vesicular appendages (Fig. 8a–g) (Supplementary Fig. 2c, d) which are formed of ectoderm, endoderm and mesoglea; the cavity of these appendages communicates with the canals of the oral arms[40]. Vesicular appendages are capable of independent movement, and during feeding of lab-reared *C. xamachana* medusae, when *Artemia* nauplii approach the oral arms, the vesicular appendage bends to cover the shrimp, thereby trapping the prey item; this trapping method was also reported in the conspecific *C. frondosa* (see ref. [32]). Clusters of cassiosomes (i.e., 30–100 individuals) line the surface of the numerous, variably sized vesicular appendages present in *C. xamachana* (Figs. 1, 8). Bigelow (1900) called these appendages nettle batteries, referring

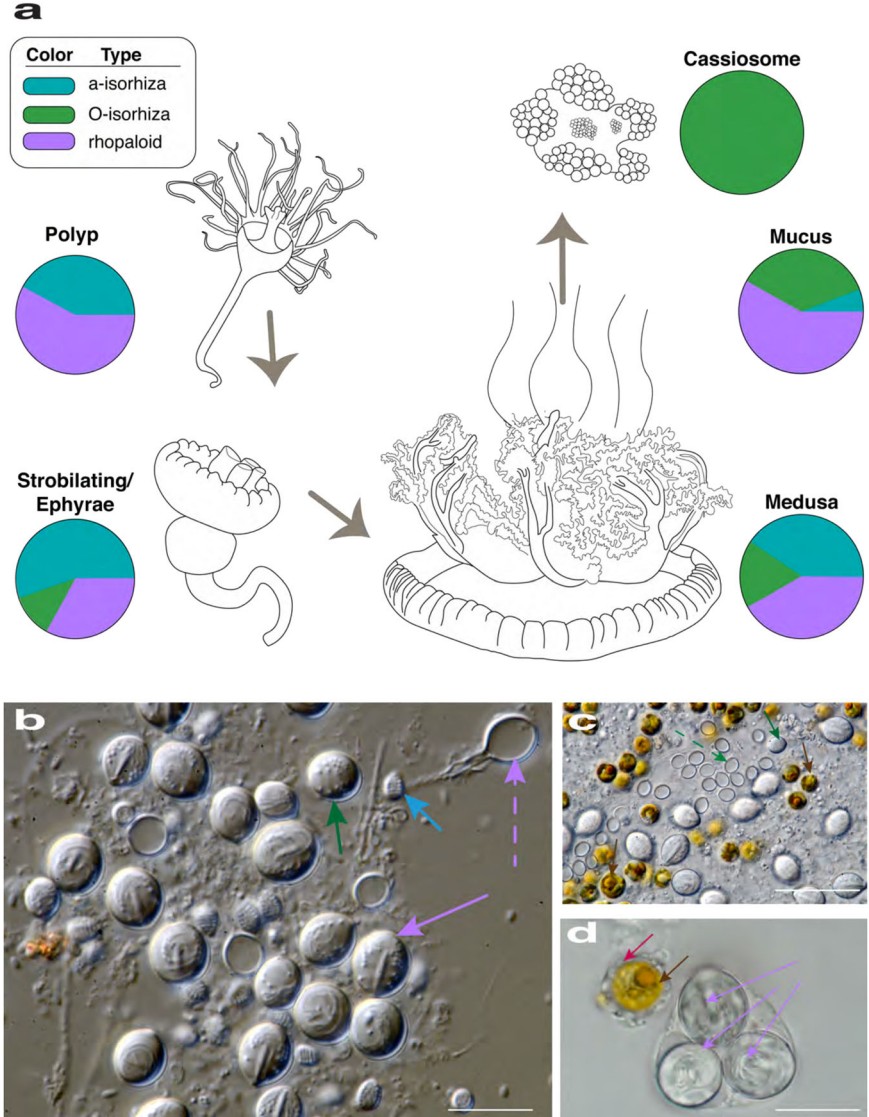

**Fig. 6 Nematocyte type proportion (cnidome) varies within different life stages and structures of *C. xamachana*. a** Figure displaying life cycle stages of *C. xamachana* (observed in this study) and associated cassiosome-laden mucus release by the medusae. Pie charts indicate proportion of nematocyte types for polyps (*n* = 3 distinct polyps), strobila/ephyrae (*n* = 3 distinct ephyrae), medusa (*n* = 3 distinct medusae), mucus (from *n* = 4 distinct medusae), and cassiosomes (from *n* = 4 distinct medusae), based on measurements of multiples of each nematocyst type per life stage (see details in Supplementary Table 1). **b, c** Different nematocyst types isolated from *C. xamachana* medusae oral-arm filaments corresponding to colors in pie charts in (**a**): a-isorhiza intact (light blue arrow), O-isorhiza intact (green arrows) and deployed (dashed green arrow), and rhopaloid intact (lavender arrow) and deployed (dashed lavender arrow); and *Symbidinium* (brown arrows). **d** Mucus contents of *C. xamachana* containing a triplet of rhopaloid nematocysts (lavender arrows) intact within nematocytes, and *Symbiodinium* (brown arrow) disassociated from jellyfish tissue but still within amoebocytes (pink arrows). Scale bar: **b, d** = 10 µm; **c** = 20 µm.

to their functional role in subduing prey, and possibly also in defense.

Images of semithin sections of five separate vesicular appendages revealed that cassiosomes develop within a depression externally on one side of a vesicular appendage, but occasionally on both sides (Figs. 8, 9). During development, cassiosomes originate proximally as protrusions of the epithelium (ectoderm) of the vesicular appendage, and then spread out distally as they develop, incorporating presumptive amoebocytes (endoderm cells that have migrated into the mesoglea), some of which host *Symbiodinium* (Figs. 8, 9). Early developing cassiosome protrusions are connected peripherally to the pocket surface of the vesicular appendage by their shared ectoderm epithelial layer, whereas fully developed cassiosomes awaiting deployment are

only loosely attached to the pocket and neighboring cassiosomes (Figs. 8, 9). This development process results in irregular popcorn-shaped cassiosomes, as shown in the 3-D reconstruction of their organization within the vesicular appendages, based on semithin images (Fig. 8).

The peripheral layer (nematocytes and ectoderm) surrounds a central space containing clusters of amoebocytes often hosting *Symbiodinium*, randomly interspersed among clear empty patches that exhibit substantially different refractive index properties (as seen in DIC) (Figs. 5, 8, 9) reminiscent of the small bags of mesoglea and nematocysts witnessed being released in conspecific *C. frondosa* by Smith (see ref. [40]). These findings corroborate those of our SEM and confocal analyses, and suggest the central region of cassiosomes is amorphous, containing only

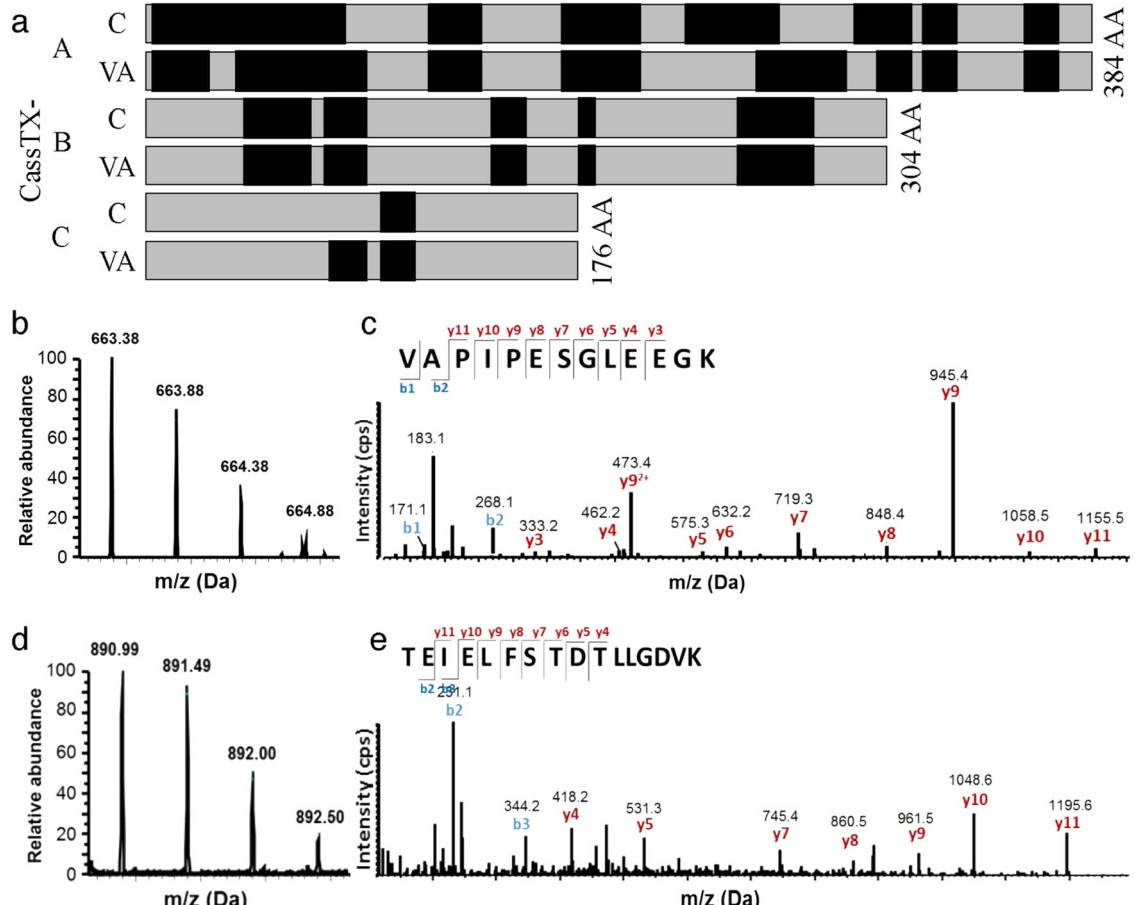

**Fig. 7 LC-MS/MS identification of jellyfish toxin proteins in cassiosomes. a** Alignment of peptides (black boxes) identified in two sample types (C = cassiosomes; VA = vesicular appendages) by shotgun proteomics to the full-length toxin homologs (CassTX-A, -B and -C; black bars) of the cnidarian-restricted CrTX/CaTX toxin family identified in *C. xamachana*. Each protein was identified with a confidence score > 0.05 and with at least two unique peptides in each sample, except for CassTX-C in the cassiosome sample. **b–e** Representative mass and annotated tandem mass spectra of CassTX-A tryptic peptides: (**b**) and (**c**) peptide sequence VAPIPESGLEEGK; (**d**) and (**e**) peptide sequence TEIELSTDLLGDVK.

some loose cells—likely amoebocytes—many of which host *Symbiodinium* (Figs. 8, 9).

**Cassiosomes in other rhizostome jellyfish species.** Jellyfish of the taxonomic order Rhizostomeae, including *Cassiopea xamachana*, all lack marginal tentacles, possessing instead oral arms covered with minute vesicular appendages. Although the main focus of this study is to provide a detailed description of cassiosomes in *C. xamachana*, in an effort to ascertain if cassiosome production is a possible apomorphy of the rhizostome jellyfish clade, we examined the mucus of additional rhizostome jellyfish taxa and documented cassiosomes in a total of four rhizostome jellyfish lineages (five different species) (Fig. 10). Cassiosomes from all six species are classified into two main types: motile, bearing cilia that propel them in the water column and non-motile, bearing no apparent motile cilia. Mucus was directly examined (using light microscopy) from three additional rhizostomes *Mastigias papua*, *Phyllorhiza punctata*, *Catostylus mosaicus* (Fig. 10d, g, m), and a single Semeastomeae (sister group) *Aurelia sp.* (Fig. 10p), all reared at the National Aquarium (Baltimore, USA). Additionally, we obtained a video from the author of a citizen scientist blog post showing abundant motile particles reportedly released by another rhizostome *Netrostoma setouchianum*, collected in Japan[41] (Fig. 10j). Although we were unable to directly examine these cellular masses from *N.*

*setouchianum* (see ref. [41]), their motility, the irregular shape they possess (Fig. 10k, l) when released from the oral arms, and the eventual loss of bumpiness and disappearance after several days matches the general description of cassiosomes we first discovered in *C. xamachana*. Cassiosomes of *M. papua* and *P. punctata* medusae are highly motile, and share the same fundamental structure, albeit exhibiting slight variations with respect to nematocyst types present within the peripheral nematocyte layer of each type (Fig. 10d–f, g–i). Superficially, *N. setouchianum* cassiosomes (Fig. 10k, l) appear to match the morphology of the two aforementioned species, however, as we were not able to examine them directly using microscopy (solely via video), the presence of associated dinoflagellates could not be confirmed. Conversely, cassiosomes in *Catostylus mosaicus* (Fig. 10n, o) exhibit several differences in that neither motility nor centralize *Symbiodinium* was observed but, rather, unidentified microalgae are distributed homogenously throughout the cell mass. No cassiosomes were found in the mucus of the semaeostome *Aurelia* sp. which lacks both vesicular appendages and endosymbiotic algae (Fig. 10p).

## Discussion
Jellyfish are remarkable aquatic animals that diverged over 600 mya and have in spite of, or possibly because of, their diplo-blastic nature, evolved a remarkable envenomation system in the

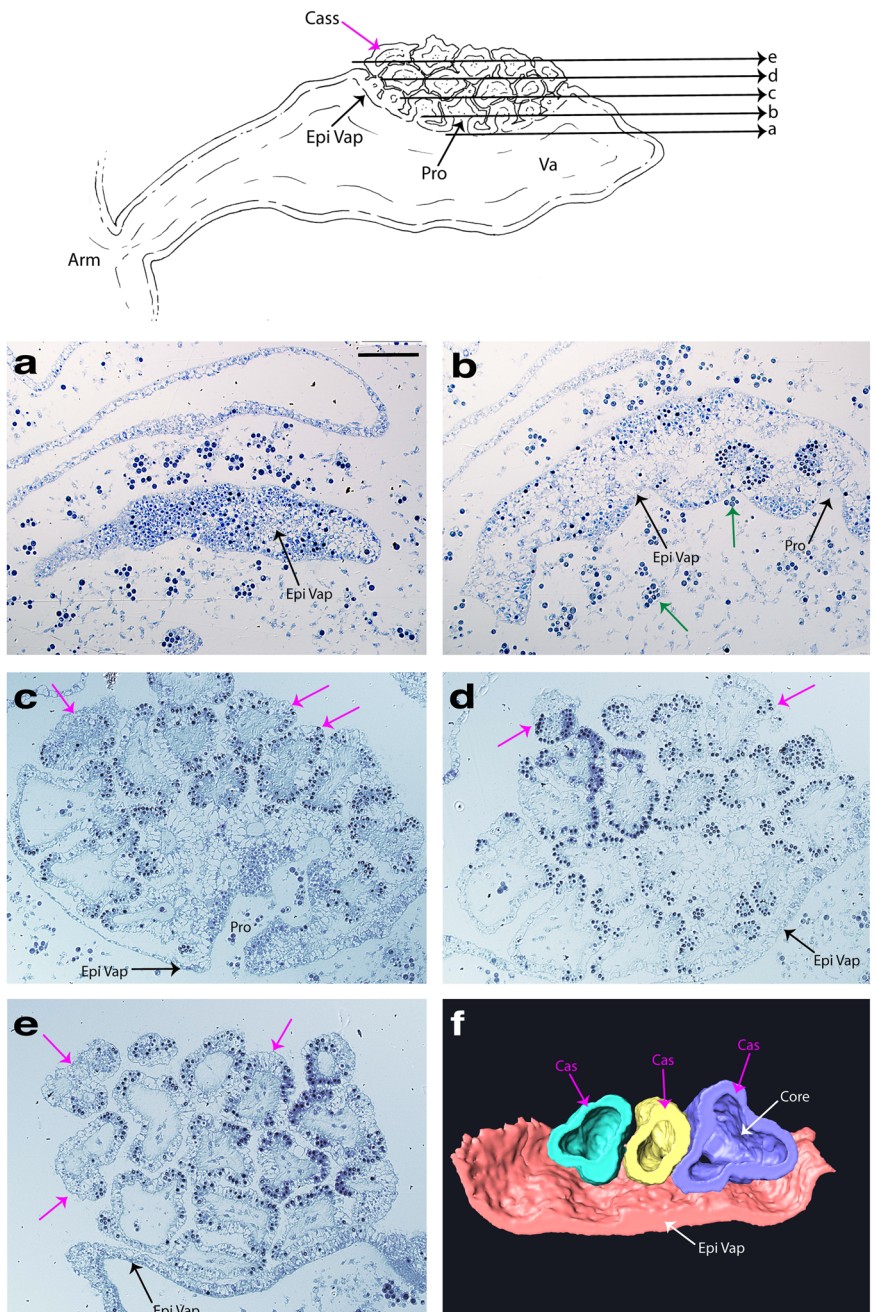

**Fig. 8 Characterization of the ultrastructure of the vesicular appendages during cassiosome production and development in *Cassiopea xamachana*.** Line drawing of vesicular appendage demonstrates how early developing cassiosome protrusions (pro) are connected peripherally to the pocket surface of the vesicular appendage by their shared epithelium (epi vap), whereas fully developed cassiosomes (cass) awaiting deployment are only loosely attached to the pocket and neighboring cassiosomes. **a–e** Semithin sections (~1 μm) of resin-embedded vesicular appendages corresponding to arrows labeled a–e in the line drawing of the vesicular appendages (va) extending from the oral arms (arm) of the medusae. Clusters of cassiosomes (pink arrows) developing from protrusions (pro) in the epithelium of the concave vesicular appendage pocket (epi vap) give rise to the cassiosome peripheral layer comprising nematocytes bearing O-isorhiza nematocysts (dark spheres stained with 1% toluidine blue) interspersed with other ectodermal cells. Clusters of amoebocytes hosting *Symbiodinium* (green arrows) move into the cassiosome core at protrusions points. Cassiosome core containing presumptive mesoglea indicated by difference in diffractive index with DIC. **f** Partial 3-D reconstruction showing protrusions developing from epithelium of the vesicular appendage pocket (epi vap) into popcorn-shaped cassiosomes. Reconstruction based on sections from a different vesicular appendage than seen above but corresponds to the region between sections a–d, revealing the empty core (core) of cassiosomes (cass) (3-D image orientation is vertical with respect to cross sections in the line drawing). arm = medusa oral arm, cass = cassiosomes(s), core = presumptive mesoglea; pro = protrusion(s); va = vesicular appendage(s), epi vap = epithelial layer of the vesicular appendage pocket. Scale bar = 250 μm.

form of stinging cells—nematocytes—for prey capture and defense. In this work, we reported the findings of an extensive investigation into the provenance, development and ultrastructure of cassiosomes, a newly described cnidarian stinging-cell structure. Based on these findings, we hypothesize that cassiosomes evolved within a single lineage of jellyfish, Rhizostomeae, to further weaponize the jellyfish by sequestering nematocytes (and other cells) into grenade-like structures that are freely released into the water within

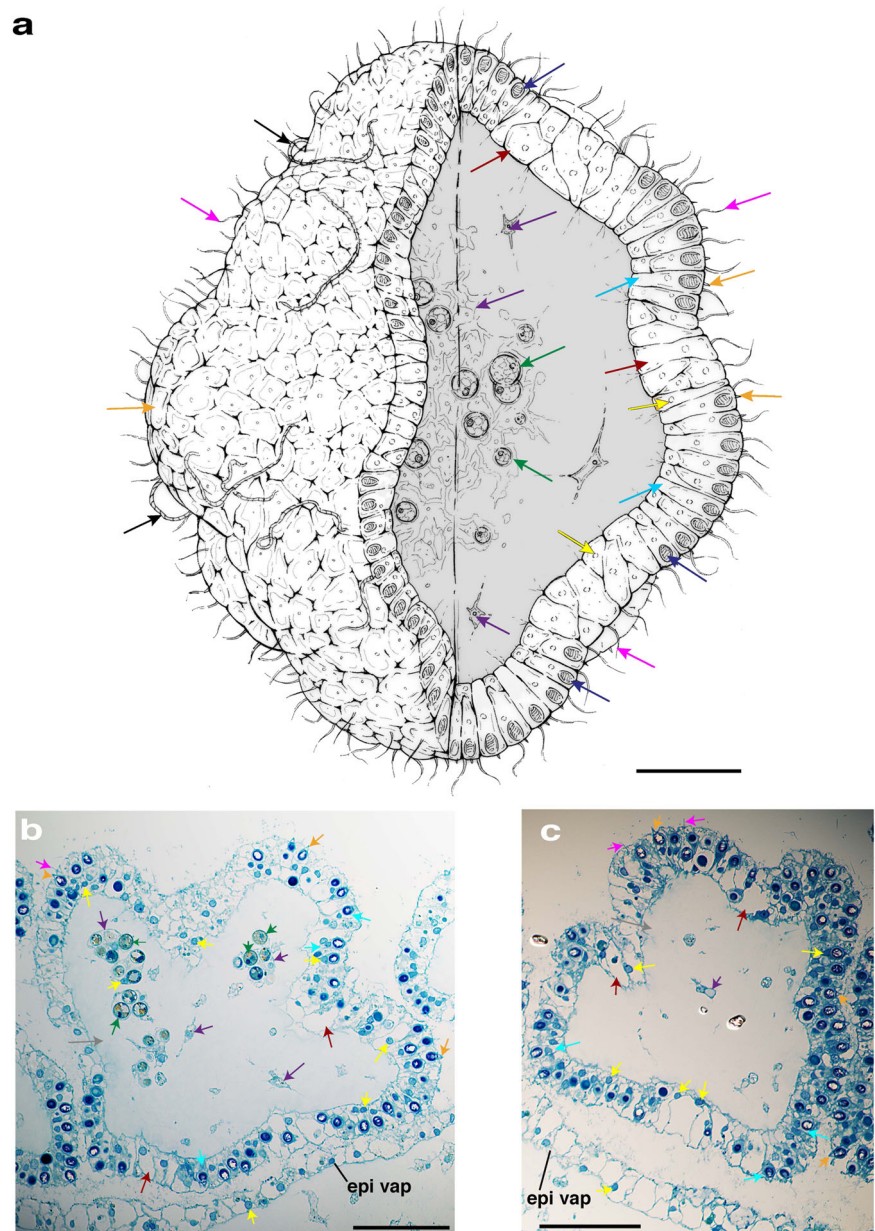

**Fig. 9 Characterization of the ultrastructure of mature cassiosomes in *Cassiopea xamachana*. a** Line drawing, and **b**, **c** thin sections of fully developed popcorn-shaped cassiosomes from semithin sections (~1 μm) of resin-embedded vesicular appendage (see Fig. 8). Cassiosome peripheral layer comprising nematocytes (cyan arrows) bearing O-isorhiza nematocysts (as peripheral dark spheres stained blue with Richardson's stain in (**b**) and (**c**)) interspersed with patches of oddly shaped ectoderm cells (red arrows), and motile cilia (pink arrows); blue-stained nuclei (yellow arrows) visible below the base of large nematocysts capsule in nematocytes, and also in non-nematocyte ectoderm cells. Cassiosome core containing presumptive mesoglea (gray central region in (**a**), gray arrow in (**b** and **c**)), speckled with amoebocytes (purple arrows)—hosting *Symbiodinium* (green arrows) or empty. Rigid stereocillia/cnidocil complex (orange arrows) visible as a point at the nematocyst apical portion, and deployed tubules (black arrows in (**a**)) on surface present as long, thick spiny threads. epi vap = epithelial layer of the vesicular appendage pocket. Scale bars = 50 μm.

exuded mucus. Our findings strongly implicate cassiosomes as a major contributor to the stinging water phenomenon reported by sea bathers and aquarists when interacting with rhizostome jellyfish species.

In this inaugural study on cassiosomes, we used extensive microscopy techniques, video-documentation and microfluidics to describe these cnidarian innovations first in *C. xamachana*, and then in taxa belonging to four additional Rhizostomeae jellyfish families. Our preliminary findings suggest there are motile and non-motile types of cassiosomes among the rhizostomes we examined in this study, and that some host endosymbiotic algae

*Symbiodinium* while at least one bears microalgae instead. However, the fundamental trait distinguishing cassiosomes from nematosomes, analogous cell masses deriving from the mesenteries of the sea anemone *Nematostella*[31], is that the structure of cassiosomes is organized into a distinct outer epithelial layer surrounding a central, mostly empty, core. Further studies are needed to elucidate the role of these photosynthetic endosymbionts in cassiosomes. The complete absence of cassiosomes in the mucus released by the semeastome *Aurelia* sp. supports our theory that cassiosomes are a rhizostome evolutionary novelty. However, a comparative examination of the mucus contents

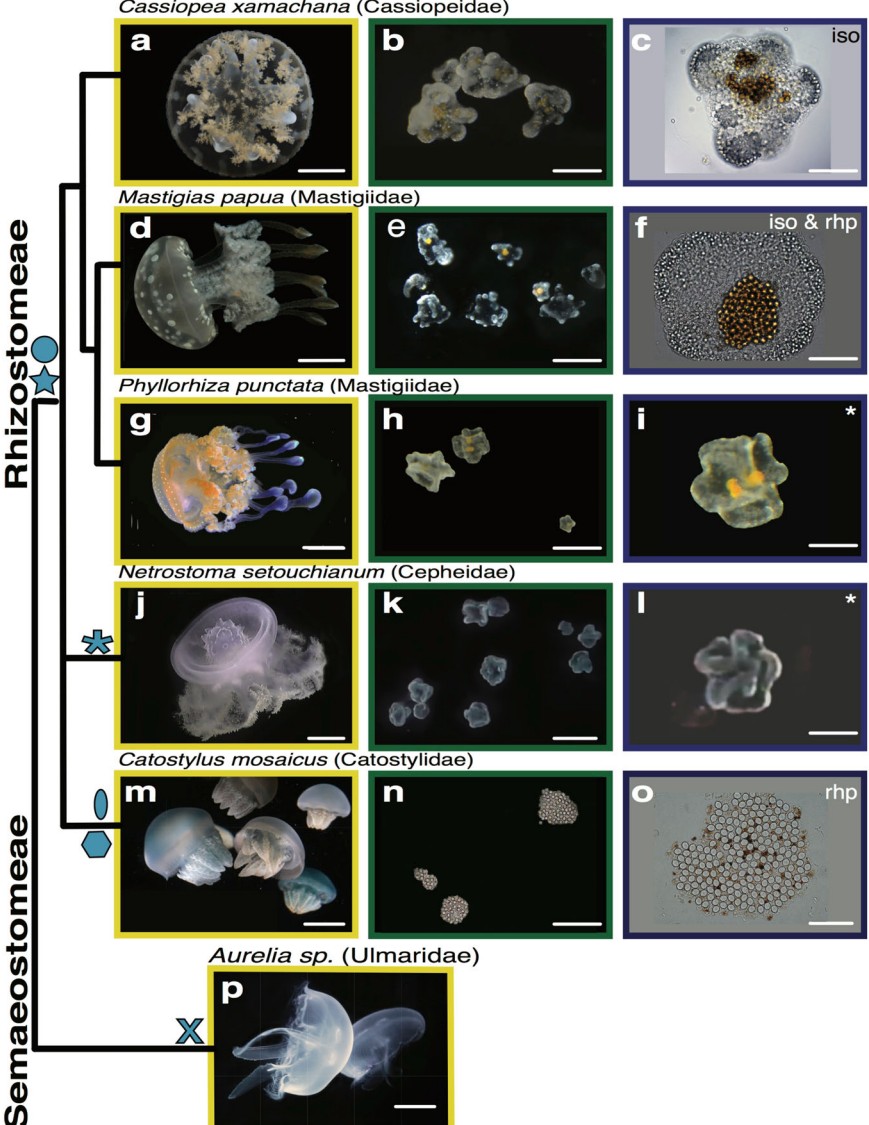

**Fig. 10 Cassiosomes observed in jellyfish species of the order Rhizostomeae.** Cladogram of species examined in this study from two orders Rhizostomeae: **a–c** *Cassiopea xamachana*, **d–f** *Mastigias papua*, **g–i** *Phyllorhiza* punctata, **j–l** *Netrostoma setouchianum*, and **m–o** *Catostylus mosaicus*, and **p** Semaeostomeae *Aurelia sp.*, and their respective cassiosome structures, when present. Abbreviations: iso = isorhiza nematocysts; rhp = rhopaloid nematocysts; * = could not confirm type of nematocysts. Blue symbols: star = motile via cilary movement; hexagon = non-motile; circle = endosymbiotic dinoflagellates within cassiosomes confirmed; oval = microalgae on the surface of cassiosomes confirmed; asterix = could not confirm presence or absence of algal symbionts; X = no cassiosomes witnessed within the mucus. Scale bar: Scale bar: **a** = 1.5 cm; **d**, **g**, **j**, **m**, **p** = 2.5 cm; **b**, **e**, **h**, **k**, **n**, **o** = 300 μm; **c**, **i**, **l** = 200 μm; **f**, **o** = 100 μm.

across all rhizostome lineages, including the eight nominal *Cassiopea* species[26,42] is needed to test this hypothesis.

Furthermore, we identified the provenance of cassiosome production and release from oral arm vesicular appendages, corroborating earlier works suggesting these vesicles (as oral vesicles[17,40,43]) function in defense and predation. These previous works noted similar structures in the oral vesicles of *Cassiopea* species, dubbed either gray bodies, bags of nematocysts and mucous cells, minute spherical bodies, or grenades, that shot when contacted. Although those reports fell short of providing an adequate description, we are confident that the structures mentioned therein correspond to what we described herein as cassiosomes. The mechanism of cassiosome deployment may vary across different rhizostome taxa, or even between conspecifics, as according to Smith[40] when prey was provided to *Cassiopea frondosa*, an aperture opened at the tip of the vesicular

appendages releasing gray bodies (putative cassiosomes) (Supplementary Fig. 2a, b). Conversely, in this study on *C. xamachana*, upon disturbance, cassiosomes spontaneously detached from the surface of vesicular appendage pockets, which lack a terminal aperture (Supplementary Fig. 2c, d).

All jellyfish have envenomation capabilities due to bioactive proteins comprising the venom cocktail of the cnidome (i.e., repertoire of nematocysts types). The cnidarian-specific pore-forming CrTX/CaTX toxin family is one of the most potent toxin groups, and represents the main proteinaceous component of the venom of cubozoans (box jellyfish), a clade that includes species whose sting results in a deadly cardiovascular condition[37]. In this study, the presence of *C. xamachana*-specific CrTX/CaTX toxin family homologs was confirmed in cassiosomes at the DNA and protein level (i.e., CassTX), validating their expression in cassiosomes and their contribution to the stinging water phenomenon.

Although *C. xamachana* in Florida waters is considered a mild stinger, reports exist of painful human envenomation resulting in rash, vomiting, painful joints, swelling and irritation for this broadly distributed species, in addition to documentation of hemolytic and cytolytic activity in crude venom of *C. xamachana* and conspecifics[29]. Following a characterization of the cnidome in *C. xamachana* in this study, we discovered a large proportion of undeployed penetrant rhopaloid nematocysts in released mucus. These findings suggest that the toxic mucus phenomenon is due to the combined effect of cassiosomes and free, undeployed nematocysts.

Overall, the topic of jellyfish mucus is an understudied field, despite the ecological importance of mucus in antimicrobial and environmental stress protection and chemical defense[44,45], relevance of toxic bioactive compounds found within venom[46,47], and the potential importance to non-cnidarian taxa. Given the recent publication of the first reference genome for *Cassiopea xamachana*[14], a thorough investigation into the molecular pathways underlying the development and release of cassiosomes and comparisons with nematocyst-enriched structures in other cnidarians (e.g., nematosomes, acontia and acrorhagi) can now be undertaken.

## Methods

**C. xamachana live animal culture.** *C. xamachana* polyps were obtained from medusae (Supplementary Fig. 1a) cultured in National Aquarium (Baltimore, USA) and maintained in the Aquaroom wet culture room, Department of Invertebrate Zoology, National Museum of Natural History (NMNH), Smithsonian Institution (Washington D.C.). Polyps were kept in petri dishes until undergoing strobilation. Metamorphosed medusae were kept either in hanging baskets within a 55-gallon tank (also housing other invertebrates), or in a separate 20-gallon tank. All cultures were kept in Artificial Seawater (29–33 ppt) on a 12-h light-dark cycle. Cultures were fed *Artemia* nauplii three days a week, followed by regular water changes.

**Isolation and preparation of cassiosomes.** *C. xamachana* medusae were placed into 150-ml glass dishes containing 45 ml of FASW (Filtered Artificial Seawater). Transferring medusae from tank to dishes caused them to release mucus within 5 min. Medusae were further agitated (for 5–10 min) by gently pipetting water onto their oral arms using a Pasteur pipette until cassiosomes were identified within the mucus as small white flecks under a dissecting microscope (Fig. 2b). Translucent mucus was collected using a transfer pipette (Supplementary Fig. 1b) and placed in a small dish of FASW for observation (Fig. 2d), or into a 1.5 ml low-bind microcentrifuge tube for molecular analysis. In addition to containing abundant cassiosomes and nematocysts, particles found suspended within the mucus included microscopic artifacts originating from other marine life inhabiting the tank (e.g., sponge spicules, hair algae, diatoms, *Artemia* and other zooplankton, and occasionally the microscopic crawling jellyfish *Eleutheria*). Mucus and isolated cassiosomes (which sink to the bottom of the dish after 15 min.) were subsequently transferred into new dishes, microfluidic devices (Supplementary Fig. 1d), or microcentrifuge tubes. Their constant motility (spinning, linear displacement, reversal, and turning) persisted for about 10 days when kept in natural light, but diminished in size and gradually lost the irregular popcorn structure (Fig. 2b–f) after 3–5 days, eventually becoming spherical by days 8–10.

**Live medusa observations.** Live medusa observations were conducted on a Nikon Stereoscope; images were captured using a Nikon D7000 Camera and strobes as needed, and analyzed in SharpShooter3 software. Occasionally, for photo-documentation purposes, medusae were temporarily immobilized using 10% MgCl$_2$, but the reagent had no effect on the motility of isolated cassiosomes. Here we acknowledge that all images appearing in this work correctly represent the original data. Many images have been digitally enhanced to a minimal degree (e.g., auto tone, auto color and/or auto contrast functions used, and backgrounds made either solid black or white) using Adobe Photoshop software. Images were further processed for publication as plates (e.g., arrows and other labels were added) using Adobe Illustrator software.

**Cnidome preparation.** *C. xamachana* squash preps made from excised oral arm tissue in medusae, bell margin tissue in ephyrae, and tentacles in polyps, cassiosomes and mucus (Fig. 6) were examined with a Nikon Eclipse e80i compound microscope and imaged with a Nikon D7000 Camera using SharpShooter3 software. Nematocyst identification and measurements (as per[7,36]) were conducted using Fiji version of ImageJ software (Version 2.0.0-rc-68/1.52f, https://imagej.net). Mean, range and standard deviations were calculated using the R base package (https://www.R-project.org/) (Supplementary Table 1).

**Discharge assays.** To observe the effects of cassiosomes on potential prey items in relatively confined spaces, discharge assays were conducted within a microfluidic device into which *Artemia* nauplii (1–2 days old) were introduced along with cassiosomes either isolated from or still within the mucus in FASW. This 6-chambered microfluidic device, with chambers 7-mm wide × 400-µm tall × 3-mm long, modified from a version originally designed as a behavioral microarena for imaging *Hydra*, was used to immobilize large cassiosomes (~400 µm diameter) or to slow down the movement of smaller cassiosomes (<400 µm) to conduct observations over periods of up to two days. Furthermore, *Artemia* nauplii (1–3 days old) were introduced into petri dishes (100-ml) containing either isolated cassiosomes, mucus lacking cassiosomes or a FASW control. Video-documentation was conducted using Nikon Stereoscope with a Nikon D7000 Camera and Sharp Shooter3 software or at the Imaging Lab at NMNH, Smithsonian Institution (Washington D.C.) using an Olympus BX63 upright microscope, and viewed with cellSens software. All movies were trimmed using iMovie and are available as Supplementary Movies.

**Scanning electron microscopy.** Cassiosomes were imaged at the Imaging Lab at the NMNH, Smithsonian Institution (Washington D.C.), with an Apreo FESEM (FEI) Scanning Electron Microscope at 5.00 KV. Thousands of isolated cassiosomes were concentrated into 1.5 ml Eppendorf tubes and fixed in freshly mixed 2.5% glutaraldehyde in FASW (33 ppt). The following day, cassiosomes were transferred to plastic BEEM® capsules (13 mm diameter) with the bottom 1/3 cut off and replaced with a modified bottom with a 100-µm mesh filter. Samples were washed with distilled water (3×, 10 min), post-fixed in 2% osmium (60 min), washed in 1× PBS (3×, 10 min), and brought through an ethanol dehydration series, 25% (5 min), 50% (5 min), 75% (5 min), 95% (60 min), 100 (3 × 15 min). BEEM® capsules containing dehydrated cassiosomes were covered with modified lids with a 100 µm mesh filter and subjected to critical point drying. Each mesh filter containing dehydrated cassiosomes was mounted onto an SEM stub (13 mm) using an adhesive carbon strip and sputtered coated with gold-palladium.

**Confocal microscopy.** Imaging of live and fixed cassiosomes was conducted at the Naval Research Laboratory (Washington D.C.), on a Nikon A1R Confocal (Nikon Instruments) microscope using both DIC and confocal laser scanning. Highly motile cassiosomes were immobilized on MatTek glass bottom dishes coated with either Poly-L-Lysine (Sigma Aldrich) or Cell Tak adhesive (Corning), and on live tissue, NucBlue-Hoescht 33342 (1,100) (ThermoFisher) was used to stain nuclei. Images were collected with ×60 objective (oil).

Fixed material was prepared by fixing cassiosomes in 4% paraformaldehyde in phosphate buffered saline (PBS) for 1 h at 4 °C. Fixative was removed from the cassiosomes with three washes in PBS with 1% Triton-X for up to 48 h before staining. Cassiosomes were stored up to four days before processing. To identify tubulin, actin and nuclei in the tissue, fixed material was stained for confocal analysis using the following protocol. Tissue was blocked in 5% goat serum (Jackson ImmunoResearch, 005–000–001) for 1 h, and rinsed three times with 10-min washes in PBS with 1% Triton-X at RT. Tissue was incubated at 1:400 acetylated alpha Tubulin Antibody (6–11B-1) Alexa Fluor ®546 for 90 min in PBS with 1.5% goat serum. After removing, tissues were stained with ActinGreenTM ReadyProbesTM Reagent, 2 drops/ml for 15 min, and then with NucBlueTM Fixed Cell ReadyProbesTM, 2 drops/ml for 30 min. Tissues were then washed five times with 10-min washes of PBS and 0.1% Tween-20, and then mounted with 80% glycerol in PBS on glass slides for imaging. Imaging was performed with laser lines at 405, 488, 561, and 640 nm, and collected with a Plan Apo 100x objective. Each image was measured via channel series to minimize fluorescence overlap between channels. All images were analyzed with NIS-Elements AR imaging software (Nikon Co. Ltd).

**Whole mount semithin sections for histology.** Five vesicular appendages (~1.5 mm wide) full of cassiosomes ($n$ = 30–100 individuals) were dissected from medusae (~5 cm umbrella diameter). Tissue was fixed overnight in 2.5% glutaraldehyde in a 0.1 M phosphate buffer with 9% sucrose at pH of 7.4 at 4 ˚C. After three washes with the same buffer, the samples were fixed at room temperature for 2.5 h in 1% OsO4 in the phosphate buffer at 4˚C. Samples were dehydrated stepwise in ethanol and infiltrated with Spurr's resin. After curing, the resin block was cut to 995 nm thick sections using the procedure described by[48] to create ribbons. These ribbons were mounted on slides and stained with 1% toluidine blue or Richardson's stain (as per ref. [49]). Slides were analyzed using the software AMIRA ® for 3-D reconstructions (as per ref. [48]).

**Real-time PCR (qPCR) analysis.** Genomic DNA was extracted (Qiagen DNeasy Blood and Tissue kit) from *C. xamachana* cassiosomes preserved in ethanol and homogenized using a handheld biovortexer with stirring rods for microcentrifuge tubes (RPI International Corp., USA), and from medusa tissue of *C. xamachana* (target), *Aurelia* sp. and *Alatina alata* (two off-target taxa). All DNA extracts were quantified using a Qubit 2.0 fluorometer with the dsDNA HS Assay Kit. Species-specific primers were designed using PrimerQuest online software (https://www.idtdna.com/PrimerQuest/) to amplify the cnidarian-restricted toxin gene family (i.e. CaTX/CrTX) (primer sequences available in Supplementary Fig. 1e). The toxin

gene target was identified by BLAST+ (https://blast.ncbi.nlm.nih.gov/) homology search against the publicly available *C. xamachana* genome (see Data Availability section below). Approximately 2 ng of extracted DNA for each sample was used to conduct triplicate 20 μl qPCR amplification assays (KAPA SYBR® FAST qPCR Master Mix (2×) Kit) using a ViiA 7 Real-Time PCR System (ThermoFisher) at the Smithsonian Laboratories of Analytical Biology (Washington, D.C.). The following cycling conditions were used: enzyme activation 95 °C (3 min), denaturation 95 °C (2 s) and annealing extension/data acquisition 60 °C (1 min) for 35 cycles, followed by a 60 °C (30 s) final post-read stage.

**LC-MS/MS analysis.** Isolated cassiosomes and vesicular appendages in FASW (33 ppt) were pelleted (4000 rpm for 10 min) and resuspended in 20 μl total 1X NuPAGE LDS sample buffer (Thermo Fisher Scientific, Carlsbad, CA) and 10 mM dithiothreitol (DTT). Exclusively LCMS grade reagents were used. After sonication with a handheld biovortex (30 s, maximum bursts for 3 min) to break down the nematocysts capsule minicollagen double wall, and then in a sonication bath (10 min) to remove bubbles, samples were heated to 70 °C for 10 min and separated on a 10% NuPAGE Bis-Tris mini gel for 35 min at constant voltage (200 V). Proteins were stained with BioSafe Coomassie (Bio-Rad, Hercules, CA) and de-stained overnight with water. Ten bands were excised from the gel according to molecular weight (gel image available as Supplementary Fig. 4). Proteins in all bands were reduced and alkylated, and then digested in gel with sequencing grade trypsin (Promega, Madison, WI). Peptides were extracted from gel by two treatments with 2% formic acid in 50% acetonitrile and a final wash with 100% acetonitrile. All extraction fractions were concentrated by speed-vac, and then resuspended in 20 μl of 0.1% formic acid. One μl of each sample was injected into an LC-MS/MS system (U3000 LC coupled to Orbitrap Fusion Lumos mass spectrometer (Thermo Scientific, Waltham, MA)) for proteomic analysis. Resulting spectra were extracted, merged, and searched by Mascot (Matrix Science Inc., London, UK) against a database containing common standards and contaminants, i.e., trypsin, keratin, etc. (190 protein sequences). A searchable database was also created in-house of amino acid sequences corresponding to predicted open reading frames (ORFs), created using TransDecoder[69] (v.5.5.0), for the three cnidarian toxin proteins (CassTX-A, CassTX-B and CassTX-C) (Supplementary Fig. 3) expressed in the *C. xamachana* transcriptome (see Data Availability section below).

**Statistics and reproducibility.** The mean length and width, and range, of multiples of each nematocyst type comprising the cnidome of *Cassiopea xamachana*, were determined in triplicate for three separate medusae (or other specified life stage). Additionally, thousands of cassiosomes were extracted from the mucus of at least 20 individual medusae over the course of this project, and imaged using the various microscopy methods described herein. Harvested mucus and/or isolated cassiosomes from multiple medusae (more than five biological replicates) were combined for LC-MS/MS proteomics, and tissue subsamples were taken from three different medusae for genetic molecular analysis (PCR and qPCR). Five separate vesicular appendages were analyzed with histological methods. Detailed explanations provided within the Methods section are sufficient to ensure reproducibility of the experiments conducted herein.

**Reporting summary.** Further information on research design is available in the Nature Research Reporting Summary linked to this article.

## Data availability
The authors declare that all relevant data supporting the findings of this study are available within the manuscript and its Supplementary materials. Additional data are available from the corresponding authors upon request. The following datasets are publicly available:

(1) *Cassiopea xamachana* transcriptome (PMCID: PMC5932825) http://ryanlab.whitney.ufl.edu/downloads/Cnidaria_transcriptomes/

(2) *Cassiopea xamachana* genome (NCBI Accession: OLMO00000000.1)

(3) *Cassiopea xamachana* MS proteomics data (Fig. 6; Supplementary Figs. 3, 4, Supplementary Table 2) (PXD012177 and https://doi.org/10.6019/PXD012177): ProteomeXchange Consortium via the PRIDE partner repository[70].

(4) *Cassiopea xamachana* toxin protein GenBank accession numbers (Fig. 5; Supplementary Fig. 3). Nucleotide sequence data reported are available in the Third Party Annotation Section of the DDBJ/ENA/GenBank databases under the accession numbers TPA: BK010718, BK010719, BK010720.

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

## Acknowledgements

We thank the team that reared the medusae in National Aquarium and the Aquaroom of the Department of Invertebrate Zoology (NMNH) and S. Whittaker, Manager of the Imaging Lab, National Museum of Natural History, Smithsonian Institution, Washington DC. We thank A. Morandini and A. Migotto who kindly provided images respectively for Fig. 1a,b and Fig. 10c (Cnidarian Tree of Life Project), and T. Yamada for granting us permission to publish frame grabs from his *N. setouchianum* cassiosomes videos as Fig. 10j–l). The research was initiated by CLA with funds from the Animal Science and Research Committee (National Aquarium). Additionally, C.L.A., L.D.F., J.N.S., and M.M. acknowledge postdoctoral fellowships from the National Research Council's Research Associateships Program. A.M.L.K. acknowledges funding from a Lerner-Gray Memorial Grant for Research in Marine Science from the American Museum of Natural History and a Graduate Studies Summer Scholarship from the University of Kansas, as well as a Smithsonian Research Fellowship at the NMNH. This research was developed with funding from the Defense Advanced Research Projects Agency (DARPA) (J.T.R. and G.J.V.) and the National Science Foundation Enabling Discovery through Genomics Tools program (J.T.R.). The views, opinions and findings expressed are those of the authors and should not be interpreted as representing the official views or policies of the U.S. Navy, Department of Defense or the U.S. Government.

## Author contributions

K.M., M.K. and C.L.A. made initial observations on cassiosomes, conceived of the experiments and drafted the initial manuscript. Experimental design for this final study, including epifluorescence microscopy, qPCR, data collection and analysis were conducted by A.M.L.K., K.M. and C.L.A.; samples were prepared for LC-MS/MS by J.S. and C.L.A. with substantial instruction by D.L.; J.S. conducted LC-MS/MS analysis with supervision by D.L.; microfluidic devices were designed and constructed by K.B., with substantial instruction by JR, who also provided training on their use for A.M.L.K., K.M. and C.L.A.; SEM preparation and imaging was carried out by A.R. and C.L.A.; fixation, histology, and imaging of semithins and 3-D reconstructions were done by A.R.; illustrations of vesicular appendages and cassiosomes were conducted by N.B. and of *C. xamachana* life stages by A.M.L.K.; confocal microscopy was performed by L.D.F., M.M. and C.L.A.; J.D.J. provided lab-reared *Cassiopea* polyps, which were reared to medusa stage by K.M., M.K., A.M.L.K., C.L.A. and A.G.C.; J.D.J. assisted with mucus extraction from additional rhizostome species provided by J.D.J.; A.M.L.K. and C.L.A. drafted the manuscript and produced figures and movies; A.G.C., P.C. and G.J.V. supervised the project from start to finish. All authors revised and approved the final manuscript.

## Competing interests

The authors declare no competing interests.
