## [Peer Review File · Communications Biology]

Reviewers' comments:

Reviewer #1 (Remarks to the Author):

The authors describe nematocyst-bearing tissue particles that they call "cassiosomes" emanating from *Cassiopea xamanacha* medusae concomitant with stress-induced mucus release. These nematosome-like structures are produced in nests within appendages of the oral arms and are restricted to the medusa stage. While the study clearly identifies nematocysts in these cell aggregates as the cause of "stinging water" it falls short of a comprehensive biological characterization of "cassiosomes" and thus represents a preliminary, mostly descriptive analysis of the mentioned structures in isolation from the main organism, leaving many important questions to further studies:

1. How do "cassiosomes" develop in the mother tissue and which cell types contribute to their formation?
2. Can the authors rule out that these structures are simply debris resulting from the tissue turnover?
3. How is their release controlled and initiated at the physiological and morphological level?
4. Do "cassiosomes" have an active role in predation or symbiosis as speculated by the authors?

Before these questions are answered I feel that the study is not of immediate relevance for a broader readership as required by Communications Biology.

Minor points:

1. The argument for an evolutionary novelty is quite weak and only supported by a few isolated descriptions of CLS in other species.
2. The low mag pictures in Fig. 2 are not sufficient to characterize "cassiosomes" in situ. High magnification images of tissue preparations are needed.
3. The cellular composition of "cassiosomes" appears to be quite heterogenous. This needs to be clarified (see above).
4. Some "cassiosomes" appear as hollow spheres whereas others are densely packed cell aggregates. This needs some explanation.
5. Free floating "cassiosomes" are obviously not effective in envenomation. This questions their possible protective or predatory role for the medusae.
6. The caption of Suppl. Fig. 2 is insufficient. Also, the authors need to show the gel image of the preparation they used for mass spec analysis. How often was the analysis repeated? A single peptide hit for CassTX-C in "cassiosomes" is not significant and should be classified as such in Fig. 6a.
7. The introduction part is lengthy and unfocussed and should also consider the well-known appearance of floating loose jellyfish tentacles detached by tides that are still able to sting several days after their detachment.

Reviewer #2 (Remarks to the Author):

Toxic novel stinging-cell structures in the mucus of the upside-down jellyfish *Cassiopea xamachana* COMMSBIO-19-0300-T, Communications Biology

Summary

This manuscript investigates the origin of the stinging sensation that arises when water near *C. xamachana* becomes agitated. The authors describe a novel structure, the "cassiosome," which is a multicellular mass of tissue released from the oral arms of adult jellies into the water column. They use microscopy, proteomics, and behavioral assays to investigate the nature of this unusual tissue and describe similar tissues from other Rhizostomes. I like this paper and I want to see it published but it needs a lot of work. My major concerns are threefold: first, the experimental design is either poorly constructed or insufficiently explained to evaluate the results presented here. Throughout the manuscript there is either no replication (e.g., many of the cnidome assays) or poorly explained replication (e.g., the phototaxis assays). Second, the morphological description of the tissue lacks rigor; the authors need to provide higher quality images that clearly demonstrate the cellular elements they claim are present (e.g., nuclei and cnidocil). Third, the manuscript is somewhat disorganized and there is insufficient context for evaluating the novelty of these results. I have provided details regarding these general comments to improve the rigor of the work and extensive minor comments to

improve the clarity of the manuscript. I consider all of these suggestions to be compulsory unless otherwise specified.

General comments

The experimental design is insufficient. First, it's not clear why the authors chose a quantitative PCR protocol to amplify genomic DNA from different tissues. Genomic DNA doesn't vary across tissues so it doesn't tell you anything tissue-specific. Furthermore, many ORFs encode pseudogenes so amplification from gDNA doesn't tell you anything about gene expression. The amplification was also performed using *Cassiopeia*-specific primers which are extremely unlikely to work in any other species; thus, the conclusion that nothing was amplified in the other taxa is unsurprising. Additionally, there doesn't appear to be a control and the results do not seem to have been analyzed statistically. The authors need to perform qPCR in tissue-specific cDNA (made from extracted RNA) if they want to make any statement about the expression of this gene in *cassiopea*, use degenerate primers and a semi-quantitative protocol to make a statement about the expression of the orthologues across taxa, or remove this experiment from the manuscript. The proteomics data support the expression of the toxin in the *cassiosome* sample, which makes the gDNA experiment unnecessary anyway.

Second, the hypothesized role of the symbionts in the biology of the *cassiosomes* is unfounded and unsupported. The authors perform phototaxis assays to test a role for the symbionts in powering the movement of the *cassiosomes* but, as described, the assay does not adequately assess phototaxis. Brine shrimp (*Artemia*) are phototactic; the authors need to demonstrate that this assay is sufficient to detect phototaxis in *artemia* or remove this experiment. Also, it is not clear what "triplicates" were performed - three different dishes of *cassiosomes*? Were they isolated at three different times? Was this the same bowl of *cassiosomes* examined at three different times? The DCMU experiment may be worth keeping but there is not enough information to evaluate the design. How was the DCMU prepared? DMSO? Ethanol? Were animals treated with DCMU for the full 10 days? Was the solution refreshed periodically during that time?

Third, the attempts to assess synergistic release of *cassiosomes* from multiple medusae are not sufficiently described. The authors mention that forcefully directing water at the medusa is enough to elicit the release of mucus but when they attempt to assess a coordinated response it seems they agitated all medusae at the same time? How could synchrony be assayed if all animals are being stimulated? As described, this experiment doesn't add anything to the manuscript.

The morphological description is interesting but insufficient. First, the authors frequently discuss the role of the mucus in prey immobilization and/or defense, but as I understand it, the mucus is just the vehicle for the *cassiosomes*. The text needs to be adjusted throughout to clarify this. Second, the authors describe the presence of multiple cell types in the *cassiosomes* but the only images in which nuclei are labeled are too small to evaluate and, as far as I can tell, they only show nuclei for nematocytes. Figure 3 needs to be revised as most of the data in this figure are uninterpretable. None of the fluorescence looks specific but again, it's not really possible to evaluate any of the fluorescent images because they are too small and there is no corresponding DIC image for comparison. The nematocyte images in Figure 5 are lovely and all of the data in Figure 3 need to be at this magnification or higher. Importantly, the authors claim there are non-nematocyte cells in the *cassiosomes* (support cells, host cells containing symbionts) but they don't provide any clear evidence of nuclei in any of them. Finally, Figure 2 would benefit greatly from the inclusion of a line drawing. These structures (particularly the dermal pockets) are very hard to understand and none of the images make that any clearer.

The manuscript is not well organized. The introduction jumps from stinging water, to cnidarian symbioses, and then back to stinging water. It's not really clear what the symbiosis contributes to this study and there is no introduction to the numerous other strategies that cnidarians use for defense (acrorhigae, acontia, etc). Thus, there is no context for evaluating why the *cassiosomes* are so unusual. Also, *cassiosomes* have been described previously (albeit in a very cursory sense) but this isn't mentioned until the discussion. Additionally, the results and discussion sections reference individual panels of multiple figures in the same sentence forcing the reader to jump back and forth from figure to figure. Organizing the figures in a more logical way will alleviate some of this confusion. In particular, the data in Figure 5A could be added to the data in supplemental table 1 (and this could be moved out of the supplement into the main paper) and the images in 5B-D could be added to

Figure 4.

Below I provide an extensive list of other minor comments that I hope the authors will find helpful. I encourage the authors to make the edits suggested below to further clarify the paper.

Specific comments

1. Title: The thing that is novel about this is the way the nematocytes are introduced into the environment. The title should focus on the novel mechanism of delivery.
2. Line 29: "waters" not "mangroves". Mangroves are trees.
3. Line 34: they're not "extracellular" they're multicellular
4. Line 35: the only thing in this sentence that's really unique is their possession of symbionts. Nematosomes are motile by the same mechanisms as cassiosomes and if disturbed the animal will exude them outside of the body cavity. Likewise, acontiate anemones will exude their acontia when disturbed.
5. Line 62: "seafloor" not "mangrove floor"
6. Line 77: intracellular?
7. Line 88: mangrove habitats? Shallow water habitats?
8. Line 113: The ability to subdue prey outside of the medusa isn't really a unique trait – the tentacles are outside of the bell so all medusae can subdue prey outside of their "body", right? Outside of Scyphozoa there are many techniques for deploying nematocytes, the thing that is important about cassiosomes is that they can subdue prey/predators without direct contact.
9. Line 129: what do you mean by "interconnected" – like they share cytoplasmic bridges? If true, that needs to be documented as it would be very surprising.
10. Line 131: mucus does not cause envenomation.
11. Line 132: are the cassiosomes neutrally buoyant or is the mucus neutrally buoyant? What happens to the mucus when the cassiosomes drop to the bottom of the dish?
12. Line 135: the data supporting linear displacement are weak. Random motion seems to be the best description.
13. Line 139: which part disintegrated? Did the nematocytes fire spontaneously or just fall apart?
14. Line 143: Are there quantitative/time-series data to support the comment about replenishment?
15. Line 146: Why isn't there a picture of the cassiosome store in ephyra??
16. Line 160: You don't need TEM to see which cell they are on, you just need a higher mag image and a ciliary marker like acetylated tubulin.
17. Line 184: you haven't really described this "reserve" very thoroughly – how many cassiosomes are in there? Can they be released continuously for several hours or is it just mucus after the first 15 mins?
18. Lines 188: wait...how did mucus lacking cassiosomes ever subdue artemia???
19. Line 196: why 2-day old in this assay and 1-day old in the previous? Were you feeding the artemia? If not, the 2-day guys were nutrient depleted, which could artificially inflate the efficacy of your experiment. This experiment needs to be performed in freshly hatched or well fed artemia.
20. Line 200: this section is very confusing. First, it's not clear why you are trying to evaluate species boundaries in Cassiopea, why is that relevant to this paper? Also, which tissues were examined at each life stage? Or was it whole animals? You say homogenized oral arms were examined but several stages don't have oral arms. Also, where are the various types of nematocyte distributed in each of the stages?
21. Line 214: It seems you have little to no replication in figure 5? This needs to be fixed as nematocyte number can vary substantially across individuals and stages. This table is important and should be part of the main paper.
22. Line 227: where are the o-isorhizas in the medusa, other than in the cassiosomes?
23. Line 289: The hypothesis that cassiosomes have mouths is unfounded and citing a blog does not lend any additional credence to it.
24. Line 292: You need to show the data, even if they're negative. It would be nice to have a positive control assay to showing that you identified some cells that do engulf the beads.
25. Line 294: nematosomes are free-floating masses in the gastrovascular cavity. They come FROM the mesenteries, they are not IN the mesenteries. Also, the phagocytes in nematosomes are not thought to be involved in nutrient assimilation as there's no reason to believe these short-lived structures (nematosomes and cassiosomes) have any requirement for energy beyond what is generated intracellularly to drive ciliary motion.
26. Line 308: allopatric?

27. Line 324: The suggestion that the cassiosomes are "powered" (solar or otherwise) is misleading. Also, the "solar" power hypothesis is inappropriate considering you haven't demonstrated that the symbionts are inside host cells or that the cells in the cassiosome have any way of taking up nutrients that are produced by those symbionts.
28. Line 354: this is an interesting hypothesis but it seems like it could be easy to test, right? You have access to uninfected polyps, could you see if any of them become infected after being housed with an agitated medusa?
29. Line 378: stinging contents of mucus - fix.
30. Line 381: nematocysts in the mucus? Not nematocytes? How could they fire?
31. Line 387: immobilization could be for defense, not prey capture. Thus, cassiosomes need not play any role in nutrient acquisition.
32. Line 398: you characterized a single protein, yes? Is that enough to comment on the unique amino acid profile of the mucus?
33. Line 422: any concern that these other things in the mucus (including another species of cnidarian) could have contaminated your protein data?
34. Line 448: how were they added? Forcefully, by pipette? Doesn't that stimulate mucus release anyway?
35. Line 454: but not all stages have oral arms
36. Line 458: what statistics?
37. Line 479: "stains and lasers"? this doesn't seem accurate. No lasers are listed and FITC-latex beads are not a stain. Also, you don't show any data that correspond to the FITC-latex beads so either include the data (with a positive control) or leave this out. Should 1,100 be 1/100?
38. Line 493: Why weren't cnidarian cell membranes labeled with phalloidin?
39. Line 516: what are the elements in this stain supposed to identify?
40. Line 554: I've never used this technique so I can't really evaluate these methods but it seems to me you should have created an amino acid database of the entire Cxam predicted proteins, yes? Wouldn't that give you a better idea of whether there are other proteins that share some of the conserved domains in the CassTX protein? Transdecoder doesn't take that long to run on a whole transcriptome.

Figures

Fig 1: Panels g and h don't really fit in this figure. Move them to a separate figure and combine figs 1 and 3 into one thorough description of this tissue. You can't refer to the cell types populating this tissue until you show higher magnification and nuclear labeling. Also g and h are redundant, we can't see "dead" in a still photo so either show a higher mag picture with a region of the exoskeleton impaled with a nematocyst harpoon or just refer to the videos.

Fig 2: What is a wart? This is the first (only?) time that term is used. What's happening in panels c-e? These need to be brightened and much larger. E is out of context, you need a low mag image or a drawing to indicate which region of the animal this panel relates to. Also, what evidence do you have that this is a subdermal pocket? Do the appendages have endoderm too?

Fig3: panels a,b show the same thing and neither is high enough mag to evaluate the claims you make about the nuclei or the cnidocil. The "support" cells are not indicated by arrow and it's not clear where these are. Also are the nematocytes (and potentially other cells) sitting on a basement membrane? Is this a proper epithelial tissue? If so, this is a significant difference from nematosomes and should be described (because it's exciting!). Panels c, d are not acceptable. Some of the nematocytes have a gentle red haze around them, some don't. There's not enough mag to see cnidocils. Panel e - not high enough mag to see nuclei. The description of panel f doesn't make any sense - what dissociated? The nuclei? The cassiosomes? What cells are you talking about? If these are intact nematocytes why isn't the matrix labeled like it is in panel d? panels g and h need to be higher mag and you need a DIC image from the same z-stack or region so we know we're looking at. What you're pointing to in panel I is a bleb, not a cilium, and if you want to demonstrate that these are whole cells you need a nuclear stain. Remove j, it doesn't tell us anything. K - why don't you label with tubulin? L is uninformative. M, n are not useful and we can see the 3D structure in the white light and SEM images in figures 1 and 4.

Fig 4: in panel a, the insert is the same magnification as the main image and is, thus, not necessary. Panel b - whatever the blue arrows are pointing to is not clear.

Fig 5: panel S should be presented as a table if you can't accurately show the data in a bar graph. What is an oral arm "filament"? this seems to be the first time you use that term. Which ones are lemon shaped rhopaloids? The empty ones? Why no undischarged lemons?

Fig 6: when you collected vesicular appendages, were there cassiosomes inside? That could affect your results significantly and needs to be reported.

Fig 7: lovely but please make g- j much (MUCH) bigger so we can actually see what the CLS look like.

Supplement

Supp Fig 2: I don't understand this figure at all. First of all these are not alignments. Are you suggesting the proteomic analysis is detecting two different isoforms of this protein (represented by purple only and purple + blue) in the two tissues? What happens if you blast the purple and blue segments in the Cxam genome? How many proteins do you hit?

Supp Fig 3: Could you combine these line drawings with some higher magnification images of the actual oral vesicles into one figure so its clear how they relate? Did you see cassiosomes in the vesicles or in a nest at the base?

I, Leslie Babonis, have chosen to waive my anonymity.

Reviewer #3 (Remarks to the Author):

General comments:

This paper brings to light novel cnidarian structures named cassiosomes, found exclusively in rhizostome jellyfish, and responsible for the widespread unexplained phenomenon of "stinging water". The authors provide a thorough characterization of these structures integrating morphological, functional, behavioral, ecological, and molecular perspectives. Cassiosomes are motile, ciliated, isorhiza nematocyst clusters carrying symbiotic dinoflagellates released with the mucus of *Cassiopea* jellyfishes upon disturbance. Cassiosomes are likely involved in predator deterrence, and possibly involved in auxiliary prey capture. The authors propose a series of sound hypotheses about the functional implications of these structures, which they accomplish to test through a set of well-thought experiments. The claims made by the authors about the source, structure, and function of cassiosomes are well supported by the experiments and graphical evidence presented. Overall, this paper is exciting and delivers high quality documentation and description of new biological structures. The results presented here are novel and of great interest to cnidarian biologists as well as to public health researchers trying to mitigate and prevent harmful interactions between beachgoers and these jellyfish. Rhizostome jellyfish are a fascinating clade with unique structural adaptations to feed on very small plankton, such as the lack of a centralized mouth or tentacles, and the network of ciliated canals leading to hundreds of feeding pores. The revelation of cassiosomes as an additional apomorphy of this group is very exciting and hopefully will attract more people to work on this group. I believe this work will also attract a broader audience of zoologists, ecologists, and evolutionary biologists interested in the origins of phenotypic novelties, and defense mechanisms.

Like the authors, I remain skeptical towards the role of cassiosomes in prey capture until further evidence of mucus-entangled prey ingestion is revealed. It seems likely that discharge on bystanders, such as humans and planktonic crustaceans, is purely collateral and non-adaptive. While I don't think this is necessary for the completeness of this work, I wonder if cassiosome release can be stimulated by chemical cues from prey alone. This could be tested by dipping a fold of mesh with *Artemia* nauplii into the water.

Minor comments:

Line 160 - "[cilia] explaining, in part, their motility" it is unclear to me from the text what other form of locomotion they employ, if cilia are implied to be just partly responsible.

Line 270 - "self-propelled via their cilia based on endogenous messages expressed by the cnidarian

cells comprising them" mentions signaling but doesn't propose an explicit alternative hypothesis for energy source to contrast the photosynthetic-fueled hypothesis.

Figure 3(f,j), Figure 4 - Upon zooming into some of the panels the raster images appear to be in low resolution overlaid with vector-art arrows. Is there a higher resolution version of those panels available?

Figure 5a - y-axis labeled "Proportion" yet the units appear to be absolute counts.

Figure 7a - The source of the polytomic cladogram is not clear. Is that the most detailed phylogenetic resolution available? Bahia et al. 2010 shows the non-monophyletic affinities among members of the families labeled here. Using a cladogram of genera congruent with recently published molecular trees might be more revealing, though the conclusions probably won't change.

Supplementary Table 1 - Under "Nematocyst Type" column, change "Rhopalid" to "Rhopaloid" as used in the main text (rhopalids are hemipteran insects).

Reviewer #4 (Remarks to the Author):

The manuscript by Ames et al. provides description and some interpretation of irregular cell masses secreted with mucus from the jellyfish *Cassiopea xamachana*. Nematocytes, zooxanthellate endodermal cells plus an additional fourth cell type are shown to form those aggregates named as "cassiosomes". Bringing attention to those cell masses will eventually contribute to increase both knowledge on medusozoans and on dinoflagellate-cnidarian symbiosis, with the potential to stimulate further this research field. Indeed, I recognize the complexity of the manuscript with a number of diverse methodological approaches used to disclose novel information on these old-known, but too long disregarded cell masses released in the copious mucus secreted by most (maybe all?) Rhizostomeae jellyfish. Using the term Cassiosomes would be comparable to naming "*Drosophila mexicana*" a new fly species found in Mexico and later to discover that the species is indeed cosmopolitan..Actually, the Authors are already aware these structures are found across many Rhizostomeae species.

For this reason, may I suggest to adopt a different terminology?

In general, I have no doubt this preliminary work will stimulate further research leading to revision of textbook knowledge on scyphozoan medusae. However, at the same time, I cannot hide a sense of non-fulfilment of a general reader's expectations. Indeed, the work raised several questions, but many remaining unanswered or without direct response. In the manuscript, up to five times it is said that further studies are required to clarify one or another issue. I might be wrong, but the different methods (molecular biology, proteomics, bioassays, microscopy) seem to have been used following a frantic rather than ordered workplan. With 14 contributing authors, a straightforward line of connection among the multiple expertise at work is somehow lost.

Following the observation of cnidocytes in cassiosomes, *C. xamachana* genome has been used for qPCR primer design and detection in cassiosomes of a known cnidarian toxin gene. The main findings: the occurrence of *C. xamachana* DNA in cassiosomes, no amplification was obtained using cubozoan or scyphozoan jellyfish DNA template (did you use *Aurelia aurita* from North Sea?). This is not much advancement: microscopic observations already provide evidence of nucleated cells – including cnidocytes - secreted with mucus from *C. xamachana*... Further, LC-MS/MS analyses were used to provide evidence that cassiosomes have also toxins, not only toxin genes. Readers would be hardly surprised by this. Particularly, simpler and cheaper in vivo bioassays showed *Artemia* were rapidly envenomated by contact with cassiosomes, proving the occurrence of functional cnidocytes. So what?

Also, the *C. xamachana* cnidome have been compared with the published cnidome of the congeneric jellyfish *C. andromeda* across the various life stages. With reference to the main target of the investigation (cassiosomes' morphology, composition, role, mechanisms of release), and given that cassiosomes in *C. andromeda* were not investigated here, it is difficult to understand the reasons underlying such interspecific comparison. Later on, it is said this information may be used in combination with other characters to revise the taxonomic status of these two nominal species. Again, as currently presented, it seems the cnidome characterization of *C. xamachana* throughout the life cycle is not functional to the scope of the present manuscript (on cassiosomes). A kind of message in the bottle left to drift in the ocean, without "geographical coordinates" allowing the reader to track the route. Further, there is no information on cnidocyst variability at intra- and inter-population levels.

The most relevant information here is the occurrence in cassiosomes of O-isorhizas (one of the most common cnidocyte type across all cnidarian classes).

As a minor remark, in the introduction (L54-55) it is said all cnidocysts deliver toxins, but only penetrant stomocnidae inject toxins. To my knowledge, volvent desmonemes or other astomocnidae cnidos (based on adhesive or coiling mechanisms) do not deliver toxins.

At the start of the manuscript (L65) readers learn Cassiopea release mucus as a stress response- But later on, it is supposed to be a defense response, a predatory tool, and a way of transfer Symbiodinium cells.

Indeed, it is hypothesized that cassiosomes have a defensive role. However, extrusion of cassiosomes in the mucus take place after 10-15 minutes following medusa agitation, and may last for several hours following the induction. Mucus release is considered as primary defense mechanisms, with cassiosomes' extrusion as secondary defense mechanism. As discussion, this should not be part of the Results section. However, while the role of mucus as defensive tool is documented in other marine species, the hypothesis of "retarded delivery of armed weapons" (released 10-15 minutes after and for several hours after an aspecific cue) remains merely speculative. If available, examples of delayed or secondary defensive mechanisms in other species might help understanding on how/why such a putative defensive mechanism would have been selected for.

Further, it is assumed that cassiosomes play a role as predation facilitators. There is little doubt functional cnidocytes occur in the mucus as well as they are contained within the extruded cell masses. Either rhopaloids and cassiosome's O-isorhizas can envenomate and kill small prey upon vigorous direct contact. The Authors speculate about mucus threads filled by envenomated preys may facilitating feeding of the mucus/cassiosome source jellyfish, but there is no evidence about this.

Also, cassiosomes are supposed to have a role as vectors of Symbiodinium allowing cell transfer from jellyfish to primary polyps. I suggest to modify this sentence- Dinoflagellates are extruded just because they are associated with endodermal host cells. Because of this, their release in the water column is facilitated in terms of increasing frequency and abundance of dinoflagellate cells ready to move across polyps and jellyfish. Cassiosome-like vectors are not required to transfer Symbiodinium from corals to water column to corals again.

One of the main working hypothesis of this manuscript - "stinging water" in mangroves due to cassiosomes, a pillar issue as presented in the abstract and introduction - remain unanswered or discussed by a simple speculative corollary: Cassiopea jellyfish release cassiosomes with functional cnidocytes, thus "stinging waters" may be due to cassiosomes".. The impact of free rhopaloid cnidocysts is not adequately discussed.

In synthesis, this manuscript contains some original information on the structure and envenomation potential of cassiosomes, embedded with unnecessary additional data (eg qPCR, proteomics). To summarize relevant information:

- 1) Dinoflagellates and their host cells are found within the mucus-laden cell masses, together with functional cnidocytes (O-isorhizas) with the potential to envenomate and kill offered crustacean prey in laboratory experiments.
- 2) Although not clearly highlighted in the mucus, rhopaloids are most abundant cnidocysts in the mucus. These large penetrant capsules are involved in mucus stinging effect.
- 3) Cell aggregates are motile by means of jellyfish-derived ciliated cells
- 4) Cell aggregates are extruded from vesicular, club-shaped, jellyfish oral appendages
- 5) The reasons for liberation of cassiosomes still remain obscure, no clear evidence is provided to demonstrated they may have a predatory or defensive value.

Overall, I have the feeling these key information may be organized in a more straightforward, synthetic way than currently offered by this somehow lengthy manuscript.

Specific comments

L54-55 Astomocnidae nematocysts with adhesive mechanisms do not deliver toxins.

L65 Cassiopea medusae are thought to release mucus as a stress response. This would corroborate that the idea that the release of mucus from "control" jellyfish laying in dishes without additional cues are not valuable controls as the mucus production means they were stressed.

Many other scyphozoans (particularly Rhizostomatae but also Semeostomeae jellyfish) are known to release mucus as stress response.

L 117 - It is confirmed the absence of cassiosomes in Aurelia.. but there are no symbiotic microalgae associated to Aurelia..

L142 - "..appendages consisting of ectoderm, endoderm, and mesoglea (a central layer in cnidarians) " - I found the information in brackets superfluous (assuming the reader knows cnidarian bauplan) or insufficient (a non-expert reader would not understand central layer with respect to what).

L143 - replenishment of cassiosome stores within dermal pockets occur within 24 hours after removal of the vesicular appendage - Please clarify whether cassiosome nests appear before, during or after appendage regeneration.

L146 - In ephyrae, are these "cnidocyte source areas" composed by cnidoblasts or just fully differentiated cnidocytes?

L160-161 cilia protruding from cassiosomes- Any information regarding ciliated cells in vesicular appendages?

L168-170 - However, in the subsequent 10-15 min., thick strands of cassiosome-laden mucus were produced from the oral arm of *C. xamachana* (Fig. 2d) .. Evolution of defensive mechanisms should be in favor of fast responses. Can you explain?

L178-180_ Jellyfish laying in dishes are not in optimal rearing condition. How can you verify that new cassiosome release is true RESPONSE to cassiosomes in FASW, and not effect of stressful rearing conditions?

L182 - There is some anecdotal evidence that Rhizostomeae jellyfish can communicate. However, this communication may hardly depend on released mass of cells. How effective, how fast would be this type of signal? Water-soluble cues can be much faster.

L184-197 "Cassiosomes subdue" prey section. The key information here is that cnidocytes of cassiosomes have functional cnidocysts. I suggest to reduce it to a single sentence, moving much of this text as legend to the supplementary videos (2-5).

L 208 - that different proportionately - replace with differ

L233-255 As already mentioned, I believe these two approaches do not provide other information than showing functional cnidocytes occur in the "cassiosomes".

L237-238 Aurelia aurita and Alatina alata. Where did you collect these jellyfish? Are you confident with their ID?

L243-245 The main results of this section are a) validation of primer specificity (designed from *C. xamachana* genome) and b) presence of *C. xamachana* DNA in cassiosomes collected from *C. xamachana*. Is this really relevant information for this manuscript?

L246-252. By LC-MS/MS analyses, three isoforms of the target toxin family were identified both in cassiosomes and vesicular appendages. As such, this finding is provided to "validate the potential for envenomation by cassiosomes in the water column", with non-explicit reference to the "stinging water" for bathers. Unfortunately, there is no information on density of free cassiosomes in the water column or in mucus strands and therefore this hypothesis is hardly verifiable. In addition, the same hypothesis can be easily formulated by the far simpler microscopic analysis of cassiosomes.

More interestingly, combined to cnidocyst morphological analysis, the LC-MS/MS analyses suggest O-isorhizae contain all the three isoforms. If this interpretation is correct, it is worth mentioning.

L288 - no information is provided on the origin of moon jellyfish used for this study.

L345 - ..venom of *C. andromeda* from the Mediterranean. *Cassiopea andromeda* is not native species from the Mediterranean. Please clarify.

L346 - 351 *C. xamachana* as junior synonym of *C. andromeda*. This seems quite out the scope of the

manuscript and hard to understand the reasons for not having tested the hypothesis at molecular level.

L359 Rhizostomeae is the only medusozoan order in which medusa-dinoflagellate relationship has been documented in multiple taxa (Please provide reference here)

L380 In spite of abundance of large stinging rhopaloids in the mucus, it is said that cassiosomes are responsible of mangrove stinging waters. I suggest to adopt a more cautionary approach, clarifying that either cassiosomes and free rhopaloids in the mucus can concur to the prey envenomation and possibly to produce the stinging water phenomenon.

Overall, I suggest major revision.

Response to Reviewer 1 (original comments in black text, responses in **bold** text):

The authors describe nematocyst-bearing tissue particles that they call “cassiosomes” emanating from *Cassiopea xamanacha* medusae concomitant with stress-induced mucus release. These nematosome-like structures are produced in nests within appendages of the oral arms and are restricted to the medusa stage. While the study clearly identifies nematocysts in these cell aggregates as the cause of “stinging water” it falls short of a comprehensive biological characterization of “cassiosomes” and thus represents a preliminary, mostly descriptive analysis of the mentioned structures in isolation from the main organism, leaving many important questions to further studies:

1. How do “cassiosomes” develop in the mother tissue and which cell types contribute to their formation?

Response: We were also curious to better understand their development. Although, this study is a report of the discovery and qualitative description of these novel structures, we conducted additional histological sections of three separate vesicular appendages in an attempt to explain the genesis of cassiosomes within the vesicular appendages of the medusae. In doing so, we have ruled out our original hypothesis of cassiosome development occurring within the oral arms, followed by transportation to the vesicular appendages. Our findings are reported within the manuscript and below.

We have conducted additional histological work on the vesicular appendages, assembled the resulting images using 3-D reconstruction and with the addition of another jellyfish expert and artist to the list of coauthors (NB), we now include a schematic (Fig. 9) and accompanying descriptions that shed light on cassiosome development within the vesicular appendages (description in #3 below).

Imaging of thin sections of three separate vesicular appendages showed that cassiosomes typically develop within a concave pocket on one side of a vesicular appendage (but occasionally on both sides) from an “epithelial” layer at the proximal side of the vesicular appendage, and then spread out distally. Cassiosomes emerge as protrusions at multiple locations along the vesicular appendage pocket marking their genesis as clusters of pouch-like protrusions. As they develop, they incorporate *Cassiopea* endodermal cells and presumptive amoebocytes (endoderm cells hosting endosymbiotic algae that have migrated into the mesoglea). Early in development, cassiosome protrusions are connected peripherally to the pocket surface of the vesicular appendage by their shared epithelial layer, while fully developed cassiosomes awaiting deployment, are only loosely attached to the pocket and neighboring cassiosomes. This peculiar development process results in an irregular ‘popcorn-shaped’ cassiosome consisting of a distinct peripheral layer of primarily nematocysts and additional non-uniform endodermal cells, and central clusters of endosymbionts random interspersed among ‘empty’ patches that exhibit substantially different refractive index properties (as seen in DIC). In *Cassiopea* medusae, endodermal cells that phagocytize endosymbiotic algae migrate to the mesoglea and become amoebocytes; no longer attached to the endodermal layer, amoebocytes host proliferating *Symbiodinium* (Colley & Trench, 1985). Therefore, we surmise these ‘empty’ patches to be

mesoglea that has been incorporated into cassiosomes together with the amoebocytes - however, further work is needed to test this hypothesis.

2. Can the authors rule out that these structures are simply debris resulting from the tissue turnover?

Response: Yes. We demonstrated that cassiosomes are only released when disturbed by external agitation or provided prey items. Additionally, we describe cassiosomes as a potential apomorphy of the rhizostome jellyfish, recording the specific release from the vesicular appendages of five species. Furthermore, we conducted additional histological sections, the results of which indicate their genesis as the vesicular appendages (see above). Together, along with photographic and videographic documentation, these findings present strong evidence supporting our conclusion that cassiosomes are organized cell masses, released in a controlled manner from the medusae, rather than just sloughed-off tissue debris.

Herein, we formally define cassiosomes as microscopic, irregularly-shaped cellular masses whose peripheral cell layer is primarily composed of nematocysts (stinging cells); this layer encompasses a mass of endoderm cells and presumptive amoebocytes – endoderm-derived cells that migrate to the *Cassiopea* mesoglea after colonization by *Symbiodinium* (Colley & Trench, 1985). Cassiosomes are produced in the vesicular appendages of *Cassiopea xamachana* and other medusae of the order Rhizostomeae (Fig. 10). In response to prey and external disturbances, rhizostome medusae release mucus containing suspended cassiosomes into the surrounding water. In this study, we documented the presence of two main types of cassiosomes across different rhizostome jellyfish taxa: (1) motile, bearing cilia that propel them in the water column and (2) non-motile, bearing no apparent motile cilia.

3. How is their release controlled and initiated at the physiological and morphological level?

Response: We are also curious to know this. However, as this study is primarily a report of the discovery and qualitative description of these novel structures, we did not investigate the mechanisms of release on the physiological or morphological level. We trust that the publication of these findings will spur broad interest within both the cnidarian and endosymbiotic organism communities to further investigate cassiosomes and the underlying physiological and molecular networks regulating their development and release in *Cassiopea* and other rhizostomes.

4. Do “cassiosomes” have an active role in predation or symbiosis as speculated by the authors?

Response: We were also curious to know this, as many of the coauthors, first authors included, are all jellyfish experts. As such, we conducted the feeding studies described in the manuscript, and used videographic documentation to ascertain whether or not cassiosomes are involved in predation or symbiosis. As we report in our discussion, it is easier to see the direct link to cassiosome function in predation, as *Cassiopea* has a modified feeding mechanism that involves taking up nutrients through secondary mouth pores in the oral arms. Details are scarce of the physiology of this nutrient uptake. Likewise, although

dinoflagellates associated with the medusae function in providing photosynthates to the medusa as nutrients, but to what extent symbionts in cassiosomes do the same is unknown. As such we stop at speculating about the role of cassiosomes in predation, rather than concluding such a role. Studies to determine these definitive associations would require extensive field sampling, and observations *in situ*, and are beyond the scope of this report. We have tightened up the language throughout the revised manuscript to avoid confusion between our speculations and our conclusions about these newly discovered structures.

Minor points:

1. The argument for an evolutionary novelty is quite weak and only supported by a few isolated descriptions of CLS in other species.

Response: Yes, this paper originally began as a description in *Cassiopea* alone. However, as many of us are jellyfish experts and have access to captive jellyfish cultures we chose to add the additional data, which strongly suggests cassiosomes are a synapomorphy within this clade. Also, supporting our hypothesis is the fact that all species in this clade have vesicular appendages, a structure not present in other jellyfish clades (where cassiosomes appear to be absent). In the revision we added an additional species (*Phyllorhiza*) as another rhizostome bearing cassiosomes bringing the number of rhizostome species definitely bearing cassiosomes to five.

2. The low mag pictures in Fig. 2 are not sufficient to characterize “cassiosomes” *in situ*. High magnification images of tissue preparations are needed.

Response: We have provided high-definition photos in the revised manuscript.

3. The cellular composition of “cassiosomes” appears to be quite heterogenous. This needs to be clarified (see above).

Response: The basic structure of cassiosomes follows that same pattern: (1) Peripheral nematocytes, interspersed with (2) patches of endoderm and (3) endosymbiotic algae in host cells (as amoebocytes – see details above). Additionally, we emphasize that the clear regions (appearing empty) that surround the central area, varying in size and shape among and between cassiosomes released from each medusa, may correspond to mesoglea taken up during cassiosome formation.

4. Some “cassiosomes” appear as hollow spheres whereas others are densely packed cell aggregates. This needs some explanation.

Response: Generally speaking most cassiosomes (and we have seen thousands) have dark centers (ochre spheres) which correspond to endosymbiotic algae clusters (*Symbiodinium*). We agree that some cassiosomes appear hollow and that a small percentage of cassiosomes seem to lack endosymbionts. The uptake of symbionts into the cassiosomes is described within the revised manuscript and may account for why some cassiosomes appear to host more *Symbiodinium* than others. Motile cassiosomes tend to become spherical after 4 – 6

days and lose their endosymbionts, but the underlying process has not been examined herein, and all images and videos were taken from freshly released cassiosomes, so the lack of symbionts in some cassiosomes is a natural phenomenon. Additionally, non-motile cassiosomes from *Catostylus* lack centralized patches of *Symbiodinium* endosymbionts, and appear instead to have microalgae distributed throughout the cell mass instead, and lacks motility.

5. Free floating “cassiosomes” are obviously not effective in envenomation. This questions their possible protective or predatory role for the medusae.

Response: We disagree with this statement. The marine environment is filled with many free-floating organisms and for a jellyfish like *Cassiopea* which takes up suspended particles in the water for nutrients (in addition to photosynthates generated within its body by *Symbiodinium*), evolving a mechanism (mucus lined with densely-packed nematocyst structures) to immobilize prey in the water column, is an effective deployment of venom.

6. The caption of Suppl. Fig. 2 is insufficient. Also, the authors need to show the gel image of the preparation they used for mass spec analysis. How often was the analysis repeated? A single peptide hit for CassTX-C in “cassiosomes” is not significant and should be classified as such in Fig. 6a.

Response: The caption for Supplementary Fig. 2 has been changed. It is now **Supplementary Fig. 3: Alignment of peptides identified via LC-MS/MS to toxin protein sequences.** The protein sequence for each isoform of the cnidarian-restricted toxin (CassTX toxins -A, -B and -C) are shown. The peptides detected for each sample are shown below the protein sequence. The peptides identified for each sample have been color coded (red: only seen in the cassiosome sample; blue: only seen in the vesicular appendage sample; purple: seen in both the cassiosome and the vesicular appendage sample). The location of the peptides within the toxin protein is also highlighted and the percent coverage of these identified peptides is listed.

The gel image has been included as Supplemental Fig. 4. Fig. 7 (previously Fig. 6a) indicates the confidence of the toxin assignment via Mascot Score. We have added that CassTX-C in the cassiosome sample is not identified with high enough confidence to determine that the protein is present. We have also added to the Figure Legend “Score=Mascot probability score, assignment confidence > 60”.

7. The introduction part is lengthy and unfocussed and should also consider the well-known appearance of floating loose jellyfish tentacles detached by tides that are still able to sting several days after their detachment.

Response: We have shortened the introduction, and greatly altered the focus. We have mentioned within the manuscript many possible source of stinging water, and don’t deny the importance of “secondary” envenomation due to lose tentacle pieces in the water. We have included several additional examples of indirect envenomation mechanisms within the introduction. That said, *Cassiopea* medusae differ from the textbook idea of a jellyfish in many ways, including the fact that they do not have tentacles. Thus, it is unlikely that

swimmers in mangrove forest are experiencing “stinging water sensation” caused by *Cassiopea* tentacles floating in the water. Additionally, we emphasize that stinging water is felt in the proximity of the medusae when they are stirred up or picked up by swimmers, but otherwise no sensation is felt. This situation is suggestive of cassiosomes as the stinging water culprit, rather than broken pieces of tentacle from *Cassiopea* or another unseen jellyfish.

Response to Reviewer 2 (original comments in black text, responses in **bold** text):

This manuscript investigates the origin of the stinging sensation that arises when water near *C. xamachana* becomes agitated. The authors describe a novel structure, the “cassiosome,” which is a multicellular mass of tissue released from the oral arms of adult jellies into the water column. They use microscopy, proteomics, and behavioral assays to investigate the nature of this unusual tissue and describe similar tissues from other Rhizostomes. I like this paper and I want to see it published but it needs a lot of work. My major concerns are threefold: first, the experimental design is either poorly constructed or insufficiently explained to evaluate the results presented here. Throughout the manuscript there is either no replication (e.g., many of the cnidome assays) or poorly explained replication (e.g., the phototaxis assays). Second, the morphological description of the tissue lacks rigor; the authors need to provide higher quality images that clearly demonstrate the cellular elements they claim are present (e.g., nuclei and cnidocil). Third, the manuscript is somewhat disorganized and there is insufficient context for evaluating the novelty of these results. I have provided details regarding these general comments to improve the rigor of the work and extensive minor comments to improve the clarity of the manuscript. I consider all of these suggestions to be compulsory unless otherwise specified.

Response: All points are well-taken. As you will find, we have undergone an extensive revision to include these helpful suggestions.

The experimental design is insufficient. First, it’s not clear why the authors chose a quantitative PCR protocol to amplify genomic DNA from different tissues. Genomic DNA doesn’t vary across tissues so it doesn’t tell you anything tissue-specific. Furthermore, many ORFs encode pseudogenes so amplification from gDNA doesn’t tell you anything about gene expression. The amplification was also performed using *Cassiopeia*-specific primers which are extremely unlikely to work in any other species; thus, the conclusion that nothing was amplified in the other taxa is unsurprising. Additionally, there doesn’t appear to be a control and the results do not seem to have been analyzed statistically. The authors need to perform qPCR in tissue-specific cDNA (made from extracted RNA) if they want to make any statement about the expression of this gene in *cassiopea*, use degenerate primers and a semi-quantitative protocol to make a statement about the expression of the orthologues across taxa, or remove this experiment from the manuscript. The proteomics data support the expression of the toxin in the cassiosome sample, which makes the gDNA experiment unnecessary anyway.

Response: The use of qPCR was to demonstrate that the cassiosomes truly originate from *Cassiopea* in the water. In other words, we weren’t interested in gene expression in this assay, only the ability to detect a specific gene corresponding exclusively to *C. xamachana*

to dismiss the more than 100-year old assumption made by Perkins (1908) that ciliated structures (with unicellular zooxanthellae) observed in *Cassiopea mucus* were simply parasitic larvae of non-cnidarian affinity. We could have designed primers to exclusively amplify the 16S gene target (or another abundant universal gene) of *C. xamachana*, but we opted instead to design primers for a protein-coding cnidarian toxin gene specific to this species (for which genome and transcriptome data are available). Our qPCR assay (done in triplicate using several off-target species that didn't amplify) proves one thing: cassiosomes consist of tissue whose provenance is *C. xamachana*, rather than corresponding to a parasite or originating in another cnidarian species. PCR and Sanger sequencing of 16S would have demonstrated the same thing (i.e. cassiosomes originate from *C. xamachana*).

We realize that this use of qPCR in this sense is unconventional, however, it is increasingly becoming a useful method for detecting environmental DNA, and as such we stand by our experimental design for the clarified purpose.

Second, the hypothesized role of the symbionts in the biology of the cassiosomes is unfounded and unsupported. The authors perform phototaxis assays to test a role for the symbionts in powering the movement of the cassiosomes but, as described, the assay does not adequately assess phototaxis. Brine shrimp (*Artemia*) are phototactic; the authors need to demonstrate that this assay is sufficient to detect phototaxis in *artemia* or remove this experiment. Also, it is not clear what "triplicates" were performed - three different dishes of cassiosomes? Were they isolated at three different times? Was this the same bowl of cassiosomes examined at three different times? The DCMU experiment may be worth keeping but there is not enough information to evaluate the design. How was the DCMU prepared? DMSO? Ethanol? Were animals treated with DCMU for the full 10 days? Was the solution refreshed periodically during that time?

Response: We appreciate this comment and have removed these experimental results from the revised manuscript.

Third, the attempts to assess synergistic release of cassiosomes from multiple medusae are not sufficiently described. The authors mention that forcefully directing water at the medusa is enough to elicit the release of mucus but when they attempt to assess a coordinated response it seems they agitated all medusae at the same time? How could synchrony be assayed if all animals are being stimulated? As described, this experiment doesn't add anything to the manuscript.

Response: We agree that these findings lack data to support a definitive claim, and as such we have removed mention of this assay from the revised manuscript.

The morphological description is interesting but insufficient. First, the authors frequently discuss the role of the mucus in prey immobilization and/or defense, but as I understand it, the mucus is just the vehicle for the cassiosomes. The text needs to be adjusted throughout to clarify this.

Response: Yes. The mucus is a vehicle for the cassiosomes. However, the viscosity of the mucus also traps the tiny brine shrimp (Supplementary Video 3), and as such may also act

as some sort of a barrier for potential predators. As this has not been tested, we have removed this speculation from the text.

Second, the authors describe the presence of multiple cell types in the cassiosomes but the only images in which nuclei are labeled are too small to evaluate and, as far as I can tell, they only show nuclei for nematocytes.

Response: We agree with this comment, which spurred us to conduct additional experiments to better understand and describe the morphology of both the cassiosomes and the vesicular appendages in which they develop. Extensive details have been added to the revised manuscript that elucidate the organization of the cassiosomes within *Cassiopea*. Much higher resolution images have also been included.

Figure 3 needs to be revised as most of the data in this figure are uninterpretable. None of the fluorescence looks specific but again, it's not really possible to evaluate any of the fluorescent images because they are too small and there is no **corresponding DIC image for comparison**.

Response: We are in agreement. As such, we have replaced the low-resolution/magnification images with much higher resolution images, using additional labelling techniques and provide corresponding DIC images when useful (Fig. 5).

The nematocyte images in Figure 5 are lovely and all of the data in Figure 3 need to be at this magnification or higher.

Response: We have now provided high-resolution images for all figures in the revised submission.

Importantly, the authors claim there are non-nematocyte cells in the cassiosomes (support cells, host cells containing symbionts) but they don't provide any clear evidence of nuclei in any of them.

Response: We have included new images both of histological sections (of the cassiosomes and vesicular appendages) as well as confocal images that all clearly show the nucleus of the other endodermal cell types (Figures 5, 8 and 9).

Finally, Figure 2 would benefit greatly from the inclusion of a line drawing. These structures (particularly the dermal pockets) are very hard to understand and none of the images make that any clearer.

Response: We have included a line drawing and 3-D reconstruction of the cassiosomes developing within the vesicular appendages for clarity (Fig. 8 and 9).

The manuscript is not well organized. The introduction jumps from stinging water, to cnidarian symbioses, and then back to stinging water. It's not really clear what the symbiosis contributes to this study and there is no introduction to the numerous other strategies that cnidarians use for

defense (acrorhigae, acontia, etc). Thus, there is no context for evaluating why the cassiosomes are so unusual.

Response: We have taken this comment into account and organized the manuscript in a much clearer manner. We agree with the Reviewer’s point that cnidarian defense mechanisms are varied, as such we have included additional examples in the introduction of the manuscript to the order of:

“As such, cnidarians have evolved a remarkable envenomation mechanism that involves the deployment of subcellular stinging capsules called nematocysts from cnidarian-specific cells called nematocytes, which vary in size, morphology, and bioactive contents⁷⁻⁹. Sea anemones possess unique nematocyte-rich structures (e.g. acrorhagi, acontia)^{10,11} and employ strategies such as tentacle and column contraction and expansion to enhance nematocyst deployment for prey capture and protection, while in medusae (i.e. jellyfish) the first line of defense is their extendable nematocyte-laden tentacles that envenomate prey and predators they encounter in the water column¹, as well as humans participating in marine recreation. In addition to being stung directly by jellyfish, indirect stinging has also been reported. Some possible explanations for indirect jellyfish stings are through loose tentacle fragments (e.g., offshore jellyfish stings in fishermen¹²), envenomation by tiny or juvenile venomous jellyfish (e.g., Irukandji-like syndrome in United States Military combat divers¹³) or Sea Bathers Eruption caused by microscopic jellyfish (e.g., *Linuche unguiculata*¹⁴)”

We have left in the discussion of symbiosis as it is relevant to the process by which the cassiosomes are formed (described above and in the text). However, we have organized these topics more clearly, and hope that the Reviewers will see an improvement in the flow of the manuscript.

Also, cassiosomes have been described previously (albeit in a very cursory sense) but this isn’t mentioned until the discussion.

Response: We have moved to the Introduction all previous instances in the literature that we have interpreted as “cassiosome” reports.

Additionally, the results and discussion sections reference individual panels of multiple figures in the same sentence forcing the reader to jump back and forth from figure to figure. Organizing the figures in a more logical way will alleviate some of this confusion. In particular, the data in Figure 5A could be added to the data in supplemental table 1 (and this could be moved out of the supplement into the main paper) and the images in 5B-D could be added to Figure 4.

Response: We have taken this excellent advice and devised a more organized method of presenting the figures.

Specific comments

1. Title: The thing that is novel about this is the way the nematocytes are introduced into the environment. The title should focus on the novel mechanism of delivery.

Response: New title - “Cassiosomes: Novel, toxic cell structures released in the mucus of the upside-down jellyfish *Cassiopea xamachana*”

2. Line 29: “waters” not “mangroves”. Mangroves are trees.

Response: Changed to “coastal waters of mangrove forests”. As per the definition by NOAA: “Mangrove forests ... grow at tropical and subtropical latitudes near the equator ...”

(https://oceanservice.noaa.gov/education/kits/estuaries/media/supp_estuar06b_mangrove.html)

3. Line 34: they’re not “extracellular” they’re multicellular

Response: Now corrected.

4. Line 35: the only thing in this sentence that’s really unique is their possession of symbionts. Nematosomes are motile by the same mechanisms as cassiosomes and if disturbed the animal will exude them outside of the body cavity. Likewise, acontiate anemones will exude their acontia when disturbed.

Response: We have clarified the novelties of cassiosomes based on their development as such:

“1) their release into the water column within the mucus, 2) the ability to trap and kill prey as mobile “grenades” outside of the medusa, 3) their organization as an outer “epithelial” layer surrounding a mostly empty core (rather than just a solid ball of cells), and 4) the presence of centrally-located endosymbiotic *Symbiodinium* dinoflagellates, in contrast to nematosomes.”

5. Line 62: “seafloor” not “mangrove floor”

Response: Corrected as per comment in (2) above.

6. Line 77: intracellular?

Response: Changed to intracellular.

7. Line 88: mangrove habitats? Shallow water habitats?

Response: Corrected as per comment in (2) above.

8. Line 113: The ability to subdue prey outside of the medusa isn’t really a unique trait – the tentacles are outside of the bell so all medusae can subdue prey outside of their “body”, right?

Outside of Scyphozoa there are many techniques for deploying nematocytes, the thing that is important about cassiosomes is that they can subdue prey/predators without direct contact.

Response: We disagree with the analogy of tentacles subduing prey = cassiosomes subduing prey outside of the body. The medusa is defined as the entire juvenile or mature pelagic life form of a medusozoan (see Lewis Ames et al, 2018). Therefore, just as the bell/umbrella is one part of the medusa, the tentacles also belong to the medusa. Nematocysts are designed to subdue prey when the tissue type they reside in (tentacle or bell wart) comes into contact with potential prey items which are eventually inserted into the manubrium (or other mouth-like structure) by the tentacle. However, *Cassiopea* medusae (and their rhizostome kin) don't have tentacles, so they feed differently through the pores in their oral arms, though the method of their feeding is not well understood (Ruppert, Richard & Barnes, 2004). The only documented cases where intact nematocytes are found in the water column with some implications for defense against predation are those released by the medusa and associated with putative protection of embryo strands and spermatophores (e.g., Lewis & Long 2005, Garcia, Lewis Ames et al. 2018, and Lewis Ames et al. 2016).

9. Line 129: what do you mean by “interconnected” – like they share cytoplasmic bridges? If true, that needs to be documented as it would be very surprising.

Response: The wording has been modified.

10. Line 131: mucus does not cause envenomation.

Response: The wording has been modified.

11. Line 132: are the cassiosomes neutrally buoyant or is the mucus neutrally buoyant? What happens to the mucus when the cassiosomes drop to the bottom of the dish?

Response: The mucus is neutrally buoyant. Mucus stays in the water column after the cassiosomes fall. The wording has been modified.

12. Line 135: the data supporting linear displacement are weak. Random motion seems to be the best description.

Response: The wording has been modified.

13. Line 139: which part disintegrated? Did the nematocytes fire spontaneously or just fall apart?

Response: Sometimes when moved into FASW of differing salinity, cassiosome nematocysts fired spontaneously. However, for those cassiosomes that survived the initial transfer, they remained motile for up to 10 days, gradually losing their corrugated appearance after day 5 or 6 days, taking on a smooth spherical shape with a greatly reduced diameter; shortly after movement ceased, cassiosome remnants were broken down, presumably by microbial communities in the FASW.

14. Line 143: Are there quantitative/time-series data to support the comment about replenishment?

Response: We have removed mention of replenishment based on new histological findings.

15. Line 146: Why isn't there a picture of the cassiosome store in ephyra??

Response: No cassiosome “store” was seen in ephyrae (which lack oral arms and vesicular appendages). However, we have modified this wording to reflect new discoveries about development following the histological work.

16. Line 160: You don't need TEM to see which cell they are on, you just need a higher mag image and a ciliary marker like acetylated tubulin.

Response: We respectfully disagree. Even after labelling cassiosomes with actin and tubulin antibodies, imaging via confocal microscopy, and examining additional thin sections, the provenance of the cilia in these analogous structures (cassiosomes) remains unknown. This was also the case when Babonis *et al.* 2016 studied the cilia of nematosomes – though they showed the cell masses are in fact ciliated, they also could not determine to which cells the cilia belonged: “Thin sections indicate that type I ciliary cones are found at the apex of cnidocytes while cells with type II ciliary cones are often found adjacent to cnidocytes (Fig. 5f) in a cell type that shares morphological features with the previously described cnidocyte support cells...”. Additional thin sections and 3D reconstruction suggest the cilia belong to *Cassiopea* endodermal cells within the cassiosomes, as medusozoan nematocysts (unlike anthozoans) lack motile cilia (possessing only stiff cnidocils and stereocils). This additional discovery is clearly described in the text and Fig. 8.

17. Line 184: you haven't really described this “reserve” very thoroughly – how many cassiosomes are in there? Can they be released continuously for several hours or is it just mucus after the first 15 mins?

Response: In fact, it's the reverse. During the first 5-10 mins, just mucus (or mucus with sparse numbers of cassiosomes) is released. It appears to take a few minutes before small dots (corresponding to cassiosomes) are visible in the mucus. Once cassiosomes are noticeably abundant in the mucus, they become increasingly abundant, and continue to be released with the mucus, which permitted us to obtain cassiosomes from medusae for several hours thereafter using the gentle agitation technique. The number of cassiosomes within each vesicular appendage varies based on the size of the appendage, but typically reserves between 30 and 100 (depending on size) cassiosomes are seen on the numerous vesicular appendages of the *Cassiopea* medusae examined in this study (bell height approximately 3-5 cm).

18. Lines 188: wait...how did mucus lacking cassiosomes ever subdue artemia???

Response: Mucus traps rapidly moving *Artemia* like a net, as it is viscous though it doesn't readily kill zooplankton (Supplementary Video 3). The mucus may play a role in breaking them down into nutrients eventually absorbed via the odd feeding strategy of *Cassiopea*. We have changed "subdue" to "trap" and "kill".

19. Line 196: why 2-day old in this assay and 1-day old in the previous? Were you feeding the artemia? If not, the 2-day guys were nutrient depleted, which could artificially inflate the efficacy of your experiment. This experiment needs to be performed in freshly hatched or well fed artemia.

Response: Brine shrimp used in discharge assays within petri dishes (Supplementary Video 2) are 1-day old brine shrimp. From our observations, cassiosomes successfully immobilized 1-day old brine shrimp as quickly as 2-day old brine shrimp. We routinely feed jellyfish and polyps with *Artemia* nauplii that are 1–3 days old, and we fail to notice any difference in the ability for polyps, medusa or cassiosomes to subdue them. We don't feel additional experiments are needed to prove this, as we mention three accounts each of cassiosomes immobilizing both 1-day and 2-day old brine shrimps.

20. Line 200: this section is very confusing. First, it's not clear why you are trying to evaluate species boundaries in *Cassiopea*, why is that relevant to this paper?

Response: The Reviewer's point is well-taken. We have removed all mention of *C. andromeda*.

Also, which tissues were examined at each life stage? Or was it whole animals? You say homogenized oral arms were examined but several stages don't have oral arms. Also, where are the various types of nematocyte distributed in each of the stages?

Response: We have clarified the tissue types and nematocyst distribution.

21. Line 214: It seems you have little to no replication in figure 5? This needs to be fixed as nematocyte number can vary substantially across individuals and stages. This table is important and should be part of the main paper.

Response: We have provided triplicate measurements for all life stages, and reorganized the Tables and Figures into a clearer schematic. We respect the opinion of the Reviewer, but have decided to keep the Table in the Supplementary Materials due to the limitation of Figures and the importance of the photo-documentation for the manuscript. We have instead supplied a new Figure (Fig. 6) which shows the proportion of nematocyte types within polyps, strobila/ephyrae, medusae, mucus, and cassiosomes.

22. Line 227: where are the o-isorhizas in the medusa, other than in the cassiosomes?

Response: The Fig. 5 header states "Different nematocyst types isolated from *C. xamachana* medusae oral-arm filaments, a-isorhiza (pink arrow), O-isorhiza (white arrows)..." and we have also added this to the main text for clarity.

23. Line 289: The hypothesis that cassiosomes have mouths is unfounded and citing a blog does not lend any additional credence to it.

Response: Although we are strong supporters of citizen scientists, and the extensive information provided in blogs such as the one we cite here, the image did come from a blog where no imperial data was provided with respect to the mention of a putative mouth. As such, we have removed any mention of it from the main text.

24. Line 292: You need to show the data, even if they're negative. It would be nice to have a positive control assay to showing that you identified some cells that do engulf the beads.

Response: We removed this assay from the manuscript, as no cells engulfed the beads and as such, we have removed any mention of it from the main text.

25. Line 294: nematosomes are free-floating masses in the gastrovascular cavity. They come FROM the mesenteries, they are not IN the mesenteries. Also, the phagocytes in nematosomes are not thought to be involved in nutrient assimilation as there's no reason to believe these short-lived structures (nematosomes and cassiosomes) have any requirement for energy beyond what is generated intracellularly to drive ciliary motion.

Response: We have corrected the origin and removed the mention of nutrient assimilation.

26. Line 308: allopatric?

Response: This has been removed from the text.

27. Line 324: The suggestion that the cassiosomes are "powered" (solar or otherwise) is misleading. Also, the "solar" power hypothesis is inappropriate considering you haven't demonstrated that the symbionts are inside host cells or that the cells in the cassiosome have any way of taking up nutrients that are produced by those symbionts.

Response: This has been removed from the text.

28. Line 354: this is an interesting hypothesis but it seems like it could be easy to test, right? You have access to uninfected polyps, could you see if any of them become infected after being housed with an agitated medusa?

Response: We present this only as a hypothesis and the experiments to test this hypothesis go beyond the scope of this study. In the experiment proposed by the Reviewer, it would be very difficult (if not impossible) to determine whether the imminent colonization was in response to drops of seawater containing free *Symbiodinium*, or whether the cassiosomes infection and subsequent metamorphosis to medusa was in fact induced by cassiosomes alone. Just a drop of water from a tank in which *Cassiopea* medusae are reared can infect an entire polyp culture in the lab. In any event, we have removed this speculation.

29. Line 378: stinging contents of mucus - fix.

Response: This text has been fixed.

30. Line 381: nematocysts in the mucus? Not nematocytes? How could they fire?

Response: This text has been modified. Although, for clarification, fully fired nematocysts are seen being released entirely from nematocytes. As such, in addition to undeployed nematocysts (outside of the nematocyte), empty “deployed” nematocysts occur freely floating in the mucus.

31. Line 387: immobilization could be for defense, not prey capture. Thus, cassiosomes need not play any role in nutrient acquisition.

Response: Point well taken. See also comments above in 23-27.

32. Line 398: you characterized a single protein, yes? Is that enough to comment on the unique amino acid profile of the mucus?

Response: This comment was not based on the proteomic results but rather on Ducklow & Mitchell (reference #59 in the original manuscript) who reported the overall mucus composition (protein, polysaccharides, etc.) to be distinct from other coral reef cnidarians. We have removed this reference from the revised manuscript.

33. Line 422: any concern that these other things in the mucus (including another species of cnidarian) could have contaminated your protein data?

Response: No. We were very careful to extract the cassiosomes from the mucus by letting them fall to the bottom of the dish and then straining them to remove mucus and its contents. The qPCR data show specific amplification of the genomic regions encoding the *Cassiopea* toxin proteins (but not off-target species), and robust matches when searching against the *Cassiopea* genome and transcriptome suggest our results are not artefacts of contaminants. Assignment of mass spectrometry spectra to peptide sequences is extremely sensitive to changes in the amino acid sequence, and as the sequence of these toxins is unique to *C. xamachana*, mis-assignment due to contamination from another species is unlikely. Our manual confirmation of the MS1 and MS2 data of several identified peptides from the toxin confirms their assignment and presence in the cassiosomes.

34. Line 448: how were they added? Forcefully, by pipette? Doesn't that stimulate mucus release anyway?

Response: The treatments were gently added using a pipette. It didn't seem to stimulate mucus release as none was seen in the control. However, we have removed these assays from the revised manuscript (see above).

35. Line 454: but not all stages have oral arms

Response: This text has been modified.

36. Line 458: what statistics?

Response: This text has been modified.

37. Line 479: “stains and lasers”? this doesn’t seem accurate. No lasers are listed and FITC-latex beads are not a stain. Also, you don’t show any data that correspond to the FITC-latex beads so either include the data (with a positive control) or leave this out. Should 1,100 be 1/100?

Response: We have removed all mentions of FITC beads, as the negative results added nothing to the value of the manuscript. See comments above too.

38. Line 493: Why weren’t cnidarian cell membranes labeled with phalloidin?

Response: They are. We just don’t show that in the image. We have used new actin and tubulin markers, so this is now moot.

39. Line 516: what are the elements in this stain supposed to identify?

Response: We have provided improved sections at higher magnification that are valuable in recognizing the “clear/empty” regions of the cassiosomes that correspond to presumptive mesoglea (Fig. 5, 8 and 9).

40. Line 554: I’ve never used this technique so I can’t really evaluate these methods but it seems to me you should have created an amino acid database of the entire Cxam predicted proteins, yes? Wouldn’t that give you a better idea of whether there are other proteins that share some of the conserved domains in the CassTX protein? Transdecoder doesn’t take that long to run on a whole transcriptome.

Response: We generated mass spectrometry data corresponding to the entire cassiosome proteome (the link is provided in the manuscript to the data available on the PRIDE database). However, in this study, we were only interested in identifying the cnidarian-specific toxin CassTX. Searching against the entire genome would provide information about what other proteins are present in the cassiosomes, but not information about conserved domains shared between CassTX and other proteins. Similar proteins or genes could more appropriately and robustly be found using a different method, like comparison of the genetic information, without doing proteomics. Similarities in sequence are not noted during peptide assignment for two reasons: the mass shifts caused by changes in the amino acid sequence are large enough to be detected by the mass spectrometer, and each peptide will result in unique fragmented MS2 data. The purpose of the proteomics was to confirm that the toxins are indeed present in the cassiosomes, strengthening the argument that these structures can be the stinging substance in the water column.

Fig 1: Panels g and h don’t really fit in this figure. Move them to a separate figure and combine figs 1 and 3 into one thorough description of this tissue. You can’t refer to the cell types populating this tissue until you show higher magnification and nuclear labeling. Also g and h are

redundant, we can't see "dead" in a still photo so either show a higher mag picture with a region of the exoskeleton impaled with a nematocyst harpoon or just refer to the videos.

Response: All recommendations have made their way into the new figures (Fig. 1 and 2). In general, we appreciate the many excellent comments on our Figures. We have greatly modified the contents and order and in doing so have addressed many of the Reviewer comments.

Fig 2: What is a wart? This is the first (only?) time that term is used. What's happening in panels c-e? These need to be brightened and much larger. E is out of context, you need a low mag image or a drawing to indicate which region of the animal this panel relates to. Also, what evidence do you have that this is a subdermal pocket? Do the appendages have endoderm too?

Response: All recommendations have made their way into the new Figures. We have clarified the major terms and provided a drawing.

Fig3: panels a,b show the same thing and neither is high enough mag to evaluate the claims you make about the nuclei or the cnidocil. The "support" cells are not indicated by arrow and it's not clear where these are. Also are the nematocytes (and potentially other cells) sitting on a basement membrane? Is this a proper epithelial tissue? If so, this is a significant difference from nematosomes and should be described (because it's exciting!). Panels c, d are not acceptable. Some of the nematocytes have a gentle red haze around them, some don't. There's not enough mag to see cnidocils. Panel e – not high enough mag to see nuclei. The description of panel f doesn't make any sense – what dissociated? The nuclei? The cassiosomes? What cells are you talking about? If these are intact nematocytes why isn't the matrix labeled like it is in panel d? panels g and h need to be higher mag and you need a DIC image from the same z-stack or region so we know we're looking at. What you're pointing to in panel I is a bleb, not a cilium, and if you want to demonstrate that these are whole cells you need a nuclear stain. Remove j, it doesn't tell us anything. K – why don't you label with tubulin? L is uninformative. M, n are not useful and we can see the 3D structure in the white light and SEM images in figures 1 and 4.

Response: All advice taken. Tubulin and actin staining was conducted, and imaged along with the nuclear stain used in original Figures (Fig. 5). Additionally, we have removed many figures that added little to the paper.

Fig 4: in panel a, the insert is the same magnification as the main image and is, thus, not necessary. Panel b – whatever the blue arrows are pointing to is not clear.

Response: All advice taken. Modified and added a line drawing (Fig. 8 and 9).

Fig 5: panel S should be presented as a table if you can't accurately show the data in a bar graph. What is an oral arm "filament"? this seems to be the first time you use that term. Which ones are lemon shaped rhopaloids? The empty ones? Why no undischarged lemons?

Response: We have clarified the terms and presented the data in a clearer manner (Fig. 6).

Fig 6: when you collected vesicular appendages, were there cassiosomes inside? That could affect your results significantly and needs to be reported.

Response: Yes. They are filled with cassiosomes. We have clarified this point.

Fig 7: lovely but please make g- j much (MUCH) bigger so we can actually see what the CLS look like.

Response: All figures have been resubmitted in high resolution (Fig. 10). Details of the other species cassiosomes are not provided in this manuscript, as the focus is their discovery in *Cassiopea*. Future work will clarify the distinct characteristics of cassiosomes within other rhizostomes.

Supplement

Supp Fig 2: I don't understand this figure at all. First of all these are not alignments. Are you suggesting the proteomic analysis is detecting two different isoforms of this protein (represented by purple only and purple + blue) in the two tissues? What happens if you blast the purple and blue segments in the Cxam genome? How many proteins do you hit?

Response: We generated the database from translated genomic data for the same species of *Cassiopea*. The different colored sequences represent the peptides identified from each sample. This is just a visual representation of the portions of the toxins that were identified, showing that in many instances a large portion of the total toxin protein was identified in both samples. The absence of identified sequences between samples does not indicate that these portions of the toxins are not present in the sample, only that they were not identified by mass spectrometry. These data serve to strengthen the support that the toxins were accurately identified. We added that "Identified peptides from each sample and their alignment to the protein sequences are shown in Supplementary Fig. 2."

Supp Fig 3: Could you combine these line drawings with some higher magnification images of the actual oral vesicles into one figure so it's clear how they relate? Did you see cassiosomes in the vesicles or in a nest at the base?

Response: We have added high magnification images of the vesicular appendages and explained how our observations in *C. xamachana* differ from the observations of Smith (1936) in *C. frondosa*. Cassiosomes we seen only in "nests" on the surface of the vesicular appendages; never being deployed through an aperture.

I, Leslie Babonis, have chosen to waive my anonymity.

Response: We thank you for a such a thorough review Dr. Babonis.

Response to Reviewer 3 (original comments in black text, responses in **bold** text):

Line 160 - “[cilia] explaining, in part, their motility” it is unclear to me from the text what other form of locomotion they employ, if cilia are implied to be just partly responsible.

Response: Thank you for the opportunity to clarify this statement. We have removed “in part”, as we have shown that *Cassiopea* cassiosome motility is fully due to their cilia.

Line 270 - “self-propelled via their cilia based on endogenous messages expressed by the cnidarian cells comprising them” mentions signaling but doesn’t propose an explicit alternative hypothesis for energy source to contrast the photosynthetic-fueled hypothesis.

Response: We too are curious about what energy source may be behind their relatively long life (10 days) moving outside of the medusa. However, in the absence of data that provides any clue about endogenous cues related to cassiosome deployment, we have removed this section entirely. The point is moot with respect to this study, but awaits further investigation.

Figure 3(f,j), Figure 4 - Upon zooming into some of the panels the raster images appear to be in low resolution overlaid with vector-art arrows. Is there a higher resolution version of those panels available?

Response: All figures in the revised manuscript are now high resolution.

Figure 5a - y-axis labeled “Proportion” yet the units appear to be absolute counts.

Response: This has been corrected.

Figure 7a - The source of the polytomic cladogram is not clear. Is that the most detailed phylogenetic resolution available? Bahya et al. 2010 shows the non-monophyletic affinities among members of the families labeled here. Using a cladogram of genera congruent with recently published molecular trees might be more revealing, though the conclusions probably won’t change.

Response: This has been corrected (removed polytomy). We also added an additional rhizostome taxon that also has cassiosomes (Fig. 10).

Supplementary Table 1 - Under “Nematocyst Type” column, change “Rhopalid” to “Rhopaloid” as used in the main text (rhopalids are hemipteran insects).

Response: This has been corrected. We certainly respect the opinion of the Reviewer but have decided to keep the Table in the Supplementary Materials due to the Figure limit and the importance of the photo-documentation for the manuscript. We have supplied a new Figure (Fig. 6) to showcase the importance of nematocyte variation within the stages and tissues of *C. xamachana*.

Response to Reviewer 4 (original comments in black text, responses in **bold text**):

The manuscript by Ames et al. provides description and some interpretation of irregular cell masses secreted with mucus from the jellyfish *Cassiopea xamachana*. Nematocytes, zooxanthellate endodermal cells plus an additional fourth cell type are shown to form those aggregates named as “cassiosomes”. Bringing attention to those cell masses will eventually contribute to increase both knowledge on medusozoans and on dinoflagellate-cnidarian symbiosis, with the potential to stimulate further this research field. Indeed, I recognize the complexity of the manuscript with a number of diverse methodological approaches used to disclose novel information on these old-known, but too long disregarded cell masses released in the copious mucus secreted by most (maybe all?) Rhizostomeae jellyfish. Using the term Cassiosomes would be comparable to naming “*Drosophila mexicana*” a new fly species found in Mexico and later to discover that the species is indeed cosmopolitan. Actually, the Authors are already aware these structures are found across many Rhizostomeae species. For this reason, may I suggest to adopt a different terminology?

Response: This is a valid point. We initially termed these structures ‘cassiosomes’ in *Cassiopea* and ‘cassiosome-like structures’ in other rhizostome jellyfish. However, we revise the nomenclature to encompass the breadth of jellyfish taxa that share this character (a putative synapomorphy of Rhizostomeae jellyfish), and formally use a single term “cassiosomes” – further separated into motile and non-motile cassiosomes. While we respect the concerns of the Reviewer, we have chosen to keep cassiosomes in order to convey the original species in which these novel cell masses were discovered. That and all potential alternate terms we considered were either less informative and/or sounded similar enough to existing terminology in the jellyfish lexicon to result in confusion.

In general, I have no doubt this preliminary work will stimulate further research leading to revision of textbook knowledge on scyphozoan medusae. However, at the same time, I cannot hide a sense of non-fulfilment of a general reader’s expectations. Indeed, the work raised several questions, but many remaining unanswered or without direct response. In the manuscript, up to five times it is said that further studies are required to clarify one or another issue. I might be wrong, but the different methods (molecular biology, proteomics, bioassays, microscopy) seem to have been used following a frantic rather than ordered workplan. With 14 contributing authors, a straightforward line of connection among the multiple expertise at work is somehow lost.

Response: The point is well taken. We chose a multi-disciplinary approach to describe these stinging cell masses that are new to science. However, our efforts to appeal to the broader scientific community (i.e. readers of *Communications Biology*) resulted in information over-load and, as such, we have taken the advice of the Reviewers to narrow our focus. Based on Reviewers’ comments, we have removed a substantial number of assays which failed to significantly enhance the overall work and emphasized the ultrastructural analysis of these structures (Fig. 5, 8 and 9).

Following the observation of cnidocytes in cassiosomes, *C. xamachana* genome has been used for qPCR primer design and detection in cassiosomes of a known cnidarian toxin gene. The main

findings: the occurrence of *C. xamachana* DNA in cassiosomes, no amplification was obtained using cubozoan or semeanostome jellyfish DNA template (did you use *Aurelia aurita* from North Sea?). This is not much advancement: microscopic observations already provide evidence of nucleated cells – including cnidocytes - secreted with mucus from *C. xamachana*... Further, LC-MS/MS analyses were used to provide evidence that cassiosomes have also toxins, not only toxin genes. Readers would be hardly surprised by this. Particularly, simpler and cheaper in vivo bioassays showed *Artemia* were rapidly envenomated by contact with cassiosomes, proving the occurrence of functional cnidocytes. So what?

Response: PCR and Sanger sequencing of 16S would have demonstrated the same thing - that cassiosomes originate from *Cassiopea*, and no other organism. We realize that this use of qPCR is novel (and unconventional), however, it is increasingly becoming a useful method for studying environmental DNA. By publishing the primer set that is specific to *Cassiopea* toxins, it will be possible to identify *Cassiopea* in the field using this set of primers and environmental DNA techniques. Also, please see response above to Reviewer #2.

Also, the *C. xamachana* cnidome have been compared with the published cnidome of the congeneric jellyfish *C. andromeda* across the various life stages. With reference to the main target of the investigation (cassiosomes' morphology, composition, role, mechanisms of release), and given that cassiosomes in *C. andromeda* were not investigated here, it is difficult to understand the reasons underlying such interspecific comparison. Later on, it is said this information may be used in combination with other characters to revise the taxonomic status of these two nominal species. Again, as currently presented, it seems the cnidome characterization of *C. xamachana* throughout the life cycle is not functional to the scope of the present manuscript (on cassiosomes). A kind of message in the bottle left to drift in the ocean, without “geographical coordinates” allowing the reader to track the route. Further, there is no information on cnidocyst variability at intra- and inter-population levels. The most relevant information here is the occurrence in cassiosomes of O-isorhizas (one of the most common cnidocyte type across all cnidarian classes).

Response: The point is well taken. We have removed any reference to the potential for *C. andromeda*.

As a minor remark, in the introduction (L54-55) it is said all cnidocysts deliver toxins, but only penetrant stomocnidae inject toxins. To my knowledge, volvent desmonemes or other astomocnidae cnidos (based on adhesive or coiling mechanisms) do not deliver toxins.

Response: Yes. So-called penetrant and non-penetrant nematocysts have been described. However, we watch the artemia become penetrated by the isorhizas, and then die, suggesting the even isorhizas are capable of deploying penetrant tubules. More research is needed to address this, as venom is clearly penetrating the prey and killing it, regardless of whether we properly understand the mechanism.

At the start of the manuscript (L65) readers learn *Cassiopea* release mucus as a stress response-

But later on, it is supposed to be a defense response, a predatory tool, and a way of transfer Symbiodinium cells.

Response: The point is well taken. We speculate about the latter functions, as little to no work has been performed on this topic. We have clarified the hypothesized role of cassiosomes.

Indeed, it is hypothesized that cassiosomes have a defensive role. However, extrusion of cassiosomes in the mucus take place after 10-15 minutes following medusa agitation, and may last for several hours following the induction. Mucus release is considered as primary defense mechanisms, with cassiosomes' extrusion as secondary defense mechanism. As discussion, this should not be part of the Results section. However, while the role of mucus as defensive tool is documented in other marine species, the hypothesis of "retarded delivery of armed weapons" (released 10-15 minutes after and for several hours after an aspecific cue) remains merely speculative. If available, examples of delayed or secondary defensive mechanisms in other species might help understanding on how/why such a putative defensive mechanism would have been selected for.

Response: Good point. We have included mention of several secondary defense mechanisms using mucus in other marine animals within the Introduction.

Further, it is assumed that cassiosomes play a role as predation facilitators. There is little doubt functional cnidocytes occur in the mucus as well as they are contained within the extruded cell masses. Either rhopaloids and cassiosome's O-isorhizas can envenomate and kill small prey upon vigorous direct contact. The Authors speculate about mucus threads filled by envenomated preys may facilitating feeding of the mucus/cassiosome source jellyfish, but there is no evidence about this.

Response: The point is well taken. We have provided more information on feeding by *Cassiopea* to further support this hypothesis within the revised manuscript.

Also, cassiosomes are supposed to have a role as vectors of Symbiodinium allowing cell transfer from jellyfish to primary polyps. I suggest to modify this sentence- Dinoflagellates are extruded just because they are associated with endodermal host cells. Because of this, their release in the water column is facilitated in terms of increasing frequency and abundance of dinoflagellate cells ready to move across polyps and jellyfish. Cassiosome-like vectors are not required to transfer Symbiodinium from corals to water column to corals again.

Response: Agreed. We have removed comments related to cassiosomes as possible infection vectors.

One of the main working hypothesis of this manuscript - "stinging water" in mangroves due to cassiosomes, a pillar issue as presented in the abstract and introduction – remain unanswered or discussed by a simple speculative corollary: *Cassiopea* jellyfish release cassiosomes with functional cnidocytes, thus "stinging waters" may be due to cassiosomes". The impact of free rhopaloid cnidocysts is not adequately discussed.

Response: We have modified this section to take into account the impact of the larger penetrant nematocytes free in the mucus and clarified the putative mechanism of stinging water in the discussion and conclusion.

In synthesis, this manuscript contains some original information on the structure and envenomation potential of cassiosomes, embedded with unnecessary additional data (eg qPCR, proteomics). To summarize relevant information:

- 1) Dinoflagellates and their host cells are found within the mucus-laden cell masses, together with functional cnidocytes (O-isorhizas) with the potential to envenomate and kill offered crustacean prey in laboratory experiments.
- 2) Although not clearly highlighted in the mucus, rhopaloids are most abundant cnidocysts in the mucus. These large penetrant capsules are involved in mucus stinging effect.
- 3) Cell aggregates are motile by means of jellyfish-derived ciliated cells
- 4) Cell aggregates are extruded from vesicular, club-shaped, jellyfish oral appendages
- 5) The reasons for liberation of cassiosomes still remain obscure, no clear evidence is provided to demonstrated they may have a predatory or defensive value.

Overall, I have the feeling these key information may be organized in a more straightforward, synthetic way than currently offered by this somehow lengthy manuscript.

Response: We have shortened the manuscript and synthesized it (we feel) in a more straightforward way.

Specific comments

L54-55 Astomocnidae nematocysts with adhesive mechanisms do not deliver toxins.

Response: Yes. So-called penetrant and non-penetrant nematocysts have been described. However, we watch the artemia become penetrated by the isorhizas, and then die, suggesting the even isorhizas are capable of deploying penetrant tubules. More research is needed to address this, as venom is clearly penetrating the prey and killing it, regardless of whether we properly understand the mechanism.

L65 Cassiopea medusae are thought to release mucus as a stress response. This would corroborate that the idea that the release of mucus from “control” jellyfish laying in dishes without additional cues are not valuable controls as the mucus production means they were stressed.

Response: Some of the authors of this study have observed release of mucus naturally and felt its sting many times. Mucus release in *Cassiopea* is well-documented. Many references are now provided in the revised manuscript.

Many other scyphozoans (particularly Rhizostomateae but also Semeostomeae jellyfish) are known to release mucus as stress response.

Response: Yes. We discuss the fact that many jellyfish release mucus, even *Aurelia*. But when we sampled the mucus, we found no cassiosome-like structures.

L 117 - It is confirmed the absence of cassiosomes in *Aurelia*.. but there are no symbiotic microalgae associated to *Aurelia*.

Response: Yes. *Aurelia* has no endosymbionts (that we know), no vesicular appendages, and no reports of stinging water. That is precisely why we chose *Aurelia* (negative control) as an outgroup to support the hypothesis that cassiosomes are an apomorphy of the rhizostomes. We have also added another rhizostome species to our observations (Fig. 10).

L142 – “..appendages consisting of ectoderm, endoderm, and mesoglea (a central layer in cnidarians) “ - I found the information in brackets superfluous (assuming the reader knows cnidarian bauplan) or insufficient (a non-expert reader would not understand central layer with respect to what).

Response: This is a fair point. We suspect that many readers won't know the medusa bauplan. We have clarified this point as presumptive mesoglea has now been discovered as an important part of the cassiosomes.

L143 – replenishment of cassiosome stores within dermal pockets occur within 24 hours after removal of the vesicular appendage – Please clarify whether cassiosome nests appear before, during or after appendage regeneration.

Response: We have removed mention of these assays as no empirical evidence was generated to support this conclusion. No further work has been done to the regeneration of appendages or cassiosomes within them.

L146 - In ephyrae, are these “cnidocyte source areas” composed by cnidoblasts or just fully differentiated cnidocytes?

Response: The latter. No cassiosomes were seen in ephyrae which lack vesicular appendages. We have removed the mention of these source areas in ephyrae from the revised manuscript.

L160-161 cilia protruding from cassiosomes- Any information regarding ciliated cells in vesicular appendages?

Response: Yes. We have clarified that ciliated cells originate in the vesicular appendages containing the developing cassiosomes. Detailed description above.

L168-170 - However, in the subsequent 10-15 min., thick strands of cassiosome-laden mucus were produced from the oral arm of *C. xamachana* (Fig. 2d) .. Evolution of defensive mechanisms should be in favor of fast responses. Can you explain?

Response: As the mucus acts to trap (immobilizes zooplankton, etc. in the water column) it may be a first defense quick response. Keeping in mind of course that the waters of mangrove forests are filled with much zooplankton, so releasing cassiosomes (rather than just a mucus net) in response to every “disturbance” would potentially be a waste of resources. We have modified the speculations about mucus and its role in defense.

L178-180_ Jellyfish laying in dishes are not in optimal rearing condition. How can you verify that new cassiosome release is true RESPONSE to cassiosomes in FASW, and not effect of stressful rearing conditions?

Response: Yes. It is true that dishes are not optimal, and cannot compare to field conditions. That is the conundrum of wet-lab culture. However, the assays in this study were done in petri dishes, but the jellyfish were reared in 20 gallon aquariums in artificial seawater (from polyps originally growing in a much longer aquarium); they were fed regularly and observations were regularly conducted. The vesicular appendages are clear reservoirs for cassiosomes, release of cassiosomes from vesicular appendages was recorded in our study, and also release of cassiosomes by jellyfish reared in large aquariums at the National Aquarium (a closer to natural habitat than petri dishes). We have removed mention of the hypothesized “synergistic release” of cassiosomes instigated by the presence of other cassiosome-laden mucus in the water, as no empirical data were provided.

L182 – There is some anecdotal evidence that Rhizostomeae jellyfish can communicate. However, this communication may hardly depend on released mass of cells. How effective, how fast would be this type of signal? Water-soluble cues can be much faster.

Response: Interesting point. However, we have not looked into communication between and among rhizostomes and, therefore, cannot speculate on this topic.

L184-197 “Cassiosomes subdue” prey section. The key information here is that cnidocytes of cassiosomes have functional cnidocysts. I suggest to reduce it to a single sentence, moving much of this text as legend to the supplementary videos (2-5).

Response: Modified, as suggested.

L 208 – that different proportionately - replace with differ

Response: Modified, as suggested.

L233-255 As already mentioned, I believe these two approaches do not provide other information than showing functional cnidocytes occur in the “cassiosomes”.

Response: The videos are evidence alone that cassiosome nematocytes are functional in killing *Artemia* nauplii. We also present several images of brine shrimp impaled by cassiosomes.

L237-238 *Aurelia aurita* and *Alatina alata*. Where did you collect these jellyfish? Are you confident with their ID?

Response: The *Aurelia* observed at the Baltimore Aquarium were from multiple *Aurelia* cohabitating species, and as such the identity was not concluded. As such, we are not confident in the species-level identification of the *Aurelia* sample used in the assays. We have changed the description to include just “*Aurelia*”. We are 100% confident with the identify of *Alatina alata* which was collected by the first author (who conducted a re-description of the species – see CL Ames *et al.* 2013). We have also published sequences for several ribosomal nuclear targets in the case of *A. alata* – See Lawley, Lewis Ames *et al.* 2016).

L243-245 The main results of this section are a) validation of primer specificity (designed from *C. xamachana* genome) and b) presence of *C. xamachana* DNA in cassiosomes collected from *C. xamachana*. Is this really relevant information for this manuscript?

Response: Yes. We find this to be necessary. Please see comments above.

L246-252. By LC-MS/MS analyses, three isoforms of the target toxin family were identified both in cassiosomes and vesicular appendages. As such, this finding is provided to “validate the potential for envenomation by cassiosomes in the water column”, with non-explicit reference to the “stinging water” for bathers. Unfortunately, there is no information on density of free cassiosomes in the water column or in mucus strands and therefore this hypothesis is hardly verifiable. In addition, the same hypothesis can be easily formulated by the far simpler microscopic analysis of cassiosomes. More interestingly, combined to cnidocyst morphological analysis, the LC-MS/MS analyses suggest *O-isorhizae* contain all the three isoforms. If this interpretation is correct, it is worth mentioning.

Response: The point is well taken. We have modified this in the revised manuscript.

L288 – no information is provided on the origin of moon jellyfish used for this study.

Response: All jellyfish originated in the National Aquarium. We have kept it to genus level as origins of the samples cannot be verified at this time.

L345 – ..venom of *C. andromeda* from the Mediterranean. *Cassiopea andromeda* is not native species from the Mediterranean. Please clarify.

Response: We have modified this text and removed mention of *C. andromeda*.

L346 – 351 *C. xamachana* as junior synonym of *C. andromeda*. This seems quite out the scope of the manuscript and hard to understand the reasons for not having tested the hypothesis at molecular level.

Response: We have removed all mentions of *C. andromeda* potential to be a synonym of *C. xamachana* from the manuscript.

L359 Rhizostomeae is the only medusozoan order in which medusa-dinoflagellate relationship has been documented in multiple taxa (Please provide reference here).

Response: The reference has been added. We also mention *Linuche*, a coronate jellyfish that is unique in the order in exhibiting a medusa-dinoflagellate relationship.

L380 In spite of abundance of large stinging rhopaloids in the mucus, it is said that cassiosomes are responsible of mangrove stinging waters. I suggest to adopt a more cautionary approach, clarifying that either cassiosomes and free rhopaloids in the mucus can concur to the prey envenomation and possibly to produce the stinging water phenomenon.

Response: Good point. We have modified this wording as suggested.

Again, we thank all the Reviewers for your time and consideration.

REVIEWERS' COMMENTS:

Reviewer #1 (Remarks to the Author):

The revised manuscript is significantly improved. I appreciate in particular the effort the authors made in collecting new data on cassiosome development documented by histological sections of vesicular appendages. This clarifies the cellular origin of these structures and adds much to the manuscript. I would suggest to attempt a simple collagen staining (e.g. Sirius red) to verify in addition the assumption that the "empty patches" in cassiosomes are filled with mesoglea. Otherwise, my points have been addressed by the revision. Minor comment: the new title is overcharged with adjectives ("toxic, novel") whose omission, in my opinion, would be a plus.

Reviewer #2 (Remarks to the Author):

The authors submit a revised manuscript which nicely addresses many of the concerns raised by the three reviewers. I appreciate the amount of work that went into this revision but there is still an outstanding issue that must be addressed. While the authors have provided new "high resolution" figures, high magnification images are required to evaluate their claims about the various non-nematocyte cell types present in the cassiosomes. The confocal data provided in Figure 5 are pretty but not terribly informative because they are all presented as max projections so it is not possible to associate any of the labeled nuclei with any individual cell type (nematocyte or non-nematocyte). I encourage the authors to identify a single focal plane from their confocal z-stack that clearly shows a nucleus in a cell that lacks a nematocyst. Ideally, this cell should be labeled with phalloidin (to clearly delineate the membrane around the cell) and tubulin, to show the long cilium. Figure 8 is informative as to the method by which cassiosomes are produced from the vesicular appendages but the images are not of high enough magnification (nor are they stained properly for identifying nuclei? Can't tell.) to really see nuclei in any of the individual cells. Same for Figure 9. It seems the authors already have the data (or at least the slides) on hand, they just do a better job of presenting it.

Two other minor points: (1) The authors have indeed streamlined the writing and made the manuscript much easier to follow; however, the first 3-4 paragraphs of the "Results" section present merely a review of the literature. This is rather strange and distracting and I encourage the authors to consider moving this information to the background or discussion sections. (2) I appreciate the recognition, but I do not need to be named in the Acknowledgements. I waived my anonymity only to provide the authors an opportunity to reach out to me if they wanted further clarification or help with the revisions, since we clearly share a common interest in cnidarian novelties.

Reviewer #3 (Remarks to the Author):

I'm really glad to see that all of my comments and concerns were adequately addressed. In addition, I really like the changes to the manuscript structure, it is now more concise and their core ideas are much more accessible. The new figures add much more clarity to the descriptions in the text, and the line drawings provide a major enhancement to the accessibility for a general audience and aesthetic quality of this work.

For the final proof, I would suggest a couple minor changes:

Line 495 - Change *Aurelia aurita* for *Aurelia* sp.

Fig. 10 - I would suggest horizontally extending the cladogram to make the positions of the trait-transitions more clear.

Response to Reviewer 1 (original comments in black text, responses in **bold** text):

Reviewer #1 (Remarks to the Authors):

The revised manuscript is significantly improved. I appreciate in particular the effort the authors made in collecting new data on cassiosome development documented by histological sections of vesicular appendages. This clarifies the cellular origin of these structures and adds much to the manuscript. I would suggest to attempt a simple collagen staining (e.g. Sirius red) to verify in addition the assumption that the “empty patches” in cassiosomes are filled with mesoglea. Otherwise, my points have been addressed by the revision. Minor comment: the new title is overcharged with adjectives (“toxic, novel”) whose omission, in my opinion, would be a plus.

- We appreciate the positive feedback and request to stain the mesoglea. We are currently unable to properly hydrate the tissue to perform the mesoglea staining, as the tissue is resin embedded. We understand this limitation and will keep this helpful suggestion in mind for future similar works.

- We have taken your advice and changed the title to “Cassiosomes are stinging-cell structures in the mucus of the upside-down jellyfish *Cassiopea xamachana*”

Reviewer #2 (Remarks to the Authors):

The authors submit a revised manuscript which nicely addresses many of the concerns raised by the three reviewers. I appreciate the amount of work that went into this revision but there is still an outstanding issue that must be addressed. While the authors have provided new “high resolution” figures, high magnification images are required to evaluate their claims about the various non-nematocyte cell types present in the cassiosomes. The confocal data provided in Figure 5 are pretty but not terribly informative because they are all presented as max projections so it is not possible to associate any of the labeled nuclei with any individual cell type (nematocyte or non-nematocyte). I encourage the authors to identify a single focal plane from their confocal z-stack that clearly shows a nucleus in a cell that lacks a nematocyst. Ideally, this cell should be labeled with phalloidin (to clearly delineate the membrane around the cell) and tubulin, to show the long cilium. Figure 8 is informative as to the method by which cassiosomes are produced from the vesicular appendages but the images are not of high enough magnification (nor are they stained properly for identifying nuclei? Can’t tell.) to really see nuclei in any of the individual cells. Same for Figure 9. It seems the authors already have the data (or at least the slides) on hand, they just do a better job of presenting it.

- We appreciate the positive feedback and request to image a single focal plane using phalloidin staining. We are unable to perform more confocal imagery to satisfy this request at this time. We understand this limitation and will keep this helpful suggestion in mind for future works. We have, however, provided higher resolution and better stained images of sections in Figure 9 to more clearly show the nuclei and other details of individual cassiosomes.

Two other minor points: (1) The authors have indeed streamlined the writing and made the manuscript much easier to follow; however, the first 3-4 paragraphs of the “Results” section present merely a review of the literature. This is rather strange and distracting and I encourage the authors to consider moving this information to the background or discussion sections. (2) I appreciate the recognition, but I do not need to be named in the Acknowledgements. I waived my anonymity only to provide the authors an opportunity to reach out to me if they wanted further clarification or help with the revisions, since we clearly share a common interest in cnidarian novelties.

- We appreciate the positive feedback on this revised section.

- Regarding the overview of *Cassiopea* in the results section, we feel this is valuable background information to allow readers to understand feeding, defense and endosymbiosis in this jellyfish and how those coordinated processes relate to the novel organization of cassiosomes. It is not uncommon to provide a short overview of the study organism in a publication that deals with the ecology of an organism; such an overview was requested by a Reviewer during the first review of the manuscript. As such, we have chosen to retain this section.

- We have removed the Reviewer’s name from the Acknowledgements section.

Reviewer #3 (Remarks to the Authors):

I’m really glad to see that all of my comments and concerns were adequately addressed. In addition, I really like the changes to the manuscript structure, it is now more concise and their core ideas are much more accessible. The new figures add much more clarity to the descriptions in the text, and the line drawings provide a major enhancement to the accessibility for a general audience and aesthetic quality of this work.

For the final proof, I would suggest a couple minor changes:

Line 495 - Change *Aurelia aurita* for *Aurelia* sp.

Fig. 10 - I would suggest horizontally extending the cladogram to make the positions of the trait-transitions more clear.

- We appreciate the positive feedback on the revised manuscript.

- On Line 495 (and in several other places) – We have replaced *Aurelia aurita* with *Aurelia* sp.

- In Fig. 10 – We have horizontally extended the cladogram by a few millimeters (without greatly reducing the image sizes) to make the positions of the trait-transitions clearer.

Again, we thank you and the Reviewers for your time and consideration.